# High resolution boundary conditions of an old ice target near Dome C, Antarctica

Duncan A. Young[1], Jason L. Roberts[2,3], Catherine Ritz[4,5], Massimo Frezzotti[6], Enrica Quartini[1,7], Marie G. P. Cavitte[1,7], Carly R. Tozer[3], Daniel Steinhage[8], Stefano Urbini[9], Hugh F. J. Corr[10], Tas van Ommen[2,3], and Donald D. Blankenship[1]

[1]University of Texas Institute for Geophysics, Jackson School of Geosciences, University of Texas at Austin, Austin, Texas, USA
[2]Australian Antarctic Division, Kingston, Australia
[3]Antarctic Climate and Ecosystems CRC, Hobart, Australia
[4]CNRS, IGE (UMR5183), F-38041 Grenoble, France
[5]Univ. Grenoble Alpes, IGE (UMR5183), F-38041 Grenoble, France
[6]ENEA, Rome, Italy
[7]Department of Geological Sciences, Jackson School of Geosciences, University of Texas at Austin, Austin, Texas, USA
[8]Alfred Wegener Institute Helmholtz Centre for Polar and Marine Research, Bremerhaven, Germany
[9]Istituto Nazionale di Geofisica e Vulcanologia, Rome, Italy
[10]British Antarctic Survey, Cambridge, United Kingdom

*Correspondence to:* Duncan A. Young (duncan@ig.utexas.edu)

**Abstract.** A high resolution (1 km line spacing) aerogeophysical survey was conducted over a region near the East Antarctic Ice Sheet's Dome C that may hold a 1.5 million year old climate record. We combined new ice thickness data derived from an airborne coherent radar sounder with unpublished data that was in part unavailable for earlier compilations, and we were able to remove older data with high positional uncertainties. We generated a revised high resolution DEM to investigate the potential for an old ice record in this region, and used laser altimetry to confirm a Cryosat-2 derived DEM for inferring the glaciological state of the candidate area. By measuring the specularity content of the bed, we were able to find an additional 50 subglacial lakes near the candidate site, and by Doppler focusing the radar data, we were able to map out the roughness of the bed at length scales of hundreds of meters.

We find that the primary candidate region contains elevated rough topography interspersed with scattered subglacial lakes and some regions of smoother bed. Free subglacial water appears to be restricted from bed overlain by ice thicknesses of less than 3000 m. A site near the ice divide was selected for further investigation. The high resolution of this ice thickness dataset also allows us to explore the nature of ice thickness uncertainties in the context of radar geometry and processing.

## 1 Introduction

The oldest recovered stratigraphically intact record of Antarctic ice is located in the EPICA Dome C ice core, collected near the joint Italian-French Concordia Station in Wilkes Land, Antarctica (EPICA Community Members, 2004). The interpreted section of this ice core, which extends back to 800 ka, records the isotopic and gas imprint of eight glacial cycles with a

periodicity of ∼100 ka. Marine records of oxygen isotopes, however, reveal that prior to 800 ka ago, the global climate system was driven by shorter, lower amplitude obliquity-driven ∼40 ka cycles, with an approximately 400 ka transition between the two states. A key goal of the international ice core community is to collect a deep ice core that samples both a local climate history of Antarctica and a global record of greenhouse gas concentration going back to 1.5 Ma (Fischer et al., 2013).

5 The requirements for a stratigraphically intact ice column covering the required epoch are: (1) low accumulation, to restrict vertical thinning rates and increase temporal coverage; (2) low geothermal heat flow, to restrict basal melt rates; (3) proximity to an ice divide, to limit disturbance due to lateral flow, and simplify the altitude history of the surface; (4) limited basal roughness, in order to restrict disruption of basal ice; and (5) ice thicknesses of about 2500 m, in order to limit thermal insulation of the basal ice. Items 1 and 2 interact, as low accumulation limits the downward advection of cold surface temperatures, requiring

10 low geothermal heat flow to prevent melting. Items 3 (implying elevated ice surface height), 4 (smooth subglacial topography), and 5 (implying limited ice thickness) lead to the somewhat contradictory requirement of a flat subglacial mountain. Given the significant logistical requirements of ice core recovery, another important criterion for any old ice site is accessibility.

Based on ensemble ice sheet modeling, tuned by the then known distribution of subglacial lakes, Van Liefferinge and Pattyn (2013) identified a number of potential regions of frozen bed that might hold ice with old basal ages (Fig. 1). A key constraint

15 on this prediction was the use of the Bedmap2 ice thickness compilation (Fretwell et al., 2013), which included ice thickness data collected up to 2009. Several of the predicted sites were clustered within 50 km of Concordia Station.

The European-led Beyond EPICA group identified these sites as being of significant interest for old ice access, and requested the ongoing ICECAP (International Collaborative Exploration of the Cryosphere through Airborne Profiling, (Young et al., 2011)) project survey these sites. The follow-on US-Australian ICECAP II project was successful in conducting a systematic

20 aerogeophysical survey of the Old Ice A site (OIA) in late January 2016. This paper reports on the preliminary results of this survey.

## 2 The Dome C region

Dome C (Fig. 1) is a local topographic high in the East Antarctic Ice Sheet (EAIS), rising to 3250 m above sea level, located 1100 km from the East Antarctic coast. Dome C separates ice flowing to Totten Glacier to the northwest from ice flowing to the

25 George V Coast to the east and to Byrd Glacier to the south. A topographic saddle connects Dome C to the higher ice overlying Subglacial Lake Vostok to the southwest, through Ridge B and Dome A, the highest part of the EAIS.

### 2.1 Previous datasets

The topography of Dome C (Fig. 1) was first defined from the joint SPRI/NSF/TUD airborne surveys of the 1970s (Drewry and Jordan, 1983). These pioneering airborne radar altimetry and radar sounding observations predated GPS, and aircraft positions were constrained by pressure altimetry and inertial navigation systems with large uncertainties; however, subsequent ground-

30 based traverses and satellite radar altimetry (Bamber and Bentley, 1994) confirmed the presence of the dome. As a site of thick ice, low accumulation, and slow ice flow, it was a promising site for ice coring, with the first cores in the region acquired in

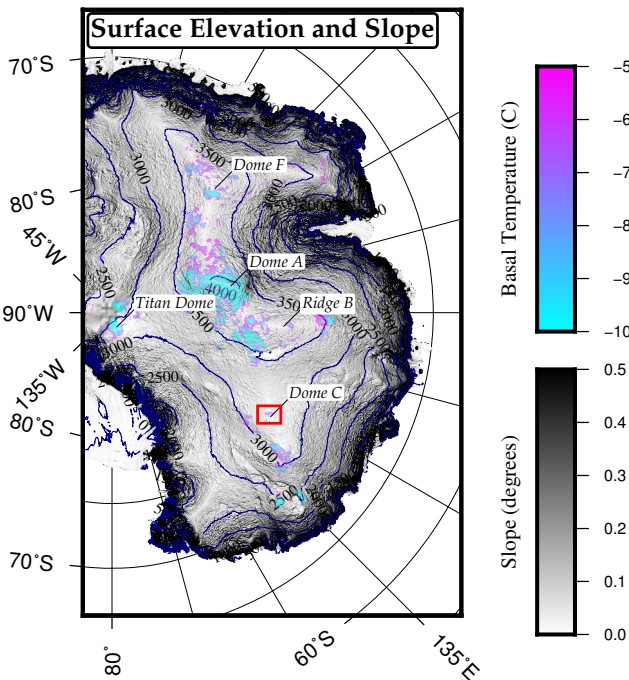

**Figure 1.** The East Antarctic ice sheet showing the Van Liefferinge and Pattyn (2013) frozen bed candidates and the location of Dome C; the region of interest is in the red box. Other potential old ice targets include Dome F, Dome A, Ridge B and Titan Dome. Surface elevation and slope are from Cryosat-2 (Helm et al., 2014). Projection is Antarctic Polar Stereographic, (EPSG:3031) with latitude of true scale at -71$^o$

1977-78 (Lorius et al., 1979). These early surveys also revealed the presence of an extensive population of subglacial lakes in this region (Oswald and Robin, 1973; Wright and Siegert, 2012).

Site selection work for the EPICA Dome C ice core took place in the mid 1990s with an Italian survey grid covering the Dome C region (Tabacco et al., 1998). This work, a combination of ground and airborne (Twin Otter) based surveys using a 60 MHz incoherent radar system with a 1 $\mu$sec pulse-width, covered most of the Dome C region with a 10 km line spacing. Ice thickness measurements from these surveys form the bulk of the data coverage for this region in the Bedmap2 compilation (Fretwell et al. (2013), see Figures 2a and 2b).

The coarse subglacial geography revealed by the Italian survey comprises of a deep subglacial trough (the Concordia Subglacial Trench) to the northeast of Dome C (see lower left of Fig. 2 a), with indications of a flat subglacial plateau near mean sea level under the center of the primary dome, and a massif to the southwest along the line of northeastward ice flow. EPICA Dome C targeted the center of the dome on the basis of apparently flat topography and its isolation from surrounding ice flow (Tabacco et al., 1998). Additional analysis (Rémy and Tabacco, 2000), however, revealed broad, shallow channels trending north-south within the subglacial plateau region.

The final EPICA Dome C ice core succeeded in obtaining ice dated as old as 800 ka; however, the lower 75 m of the ice column was either undatable, or not drilled to prevent contamination of a wet bed (Tison et al., 2015), and extrapolation

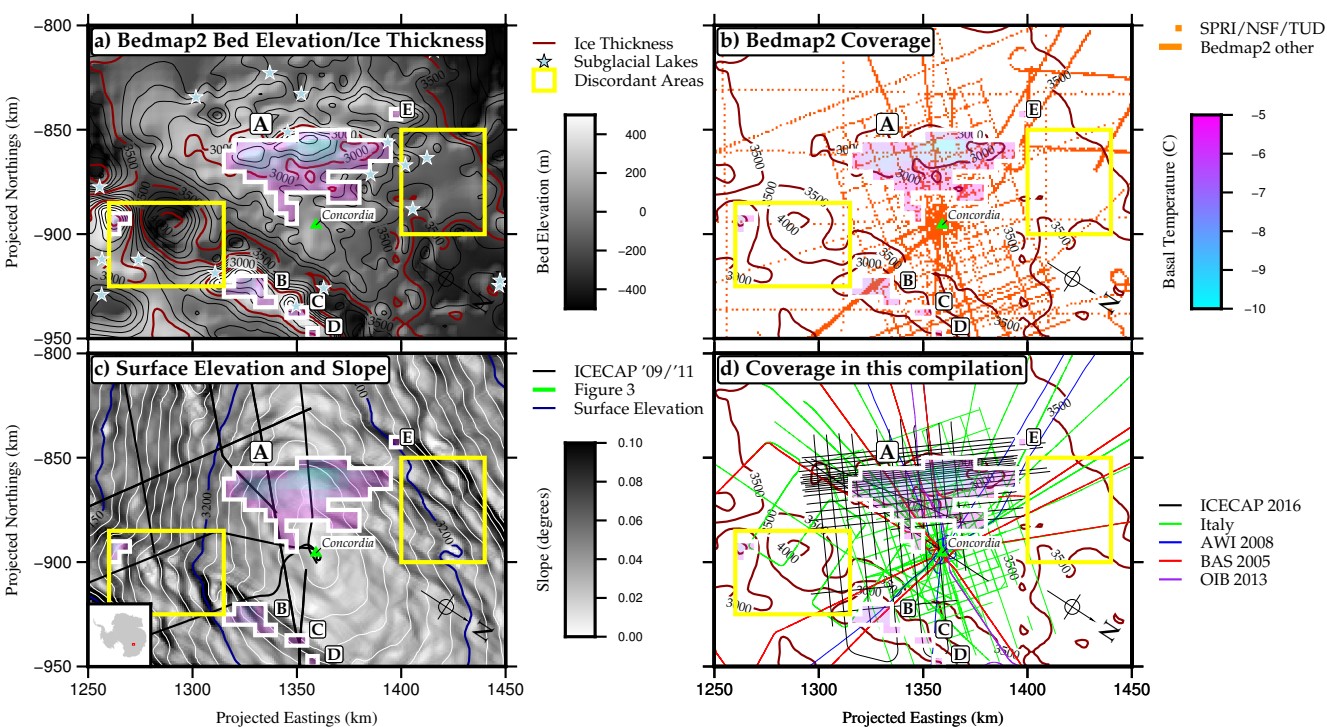

**Figure 2.** Dome C in the context of the Van Liefferinge and Pattyn (2013) frozen bed candidates and Concordia Station. A threshold of $-5°$C was used for selecting candidates. Bedmap2 bed elevation and ice thickness contours (Fretwell et al., 2013) are shown in panel a, along with subglacial lakes known as of Wright and Siegert (2012). Regions where trends seen in surface data are absent in Bedmap2 are shown by the yellow boxes in all panels. Coverage used to constrain Bedmap2 (orange) is shown in panel b, with the 1970s SPRI/NSF/TUD dataset being the sparse, dotted lines. Surface elevation and slope from Cryosat-2 (Helm et al., 2014) is shown in panel c, with older ICECAP tracks (Cavitte et al., 2016) in black and a green line showing the ice divide profile shown in Fig. 3. ICECAP Old Ice A data discussed in this paper (black), as well additional modern datasets are shown in panel d. Contours in c and d are Bedmap2 ice thickness. Projection is Antarctic Polar Stereographic, and the region corresponds to the red box in Fig. 1

of the borehole temperatures indicated that melting was likely occurring at the bed (Lefebvre et al., 2008). Analysis of the composition and structure of the lower portion of the ice core showed that focusing of ice flow by the broad channels on this plateau may have resulted in stretching and recrystallization of the lower part of the ice column, implying that an ideal old 5 ice target may require a very flat ice-bed interface around a flowline tracing back toward the ice divide, characterized by a horizontal size of several ice thicknesses (Tison et al., 2015).

In 2008, 2009, and 2011, the ICECAP project conducted survey flights using the HiCARS family of radar sounders (Young et al., 2016), mounted on a DC-3T Basler. These radar systems provided coherent, focusable 60 MHz data with a 0.08 μsec pulse-width. The goal of these flights was improving the radar stratigraphy between the EPICA Dome C and Vostok ice core sites (Cavitte et al., 2016). Included in these ICECAP flight lines was a transect along the Dome C to Subglacial Lake

Vostok ice divide, which was also flown by a range of other radar sounders, as well as a number of sparse lines of the Vostok/Concordia/Dumont d'Urville corridor (VCD), typically 20 to 40 km apart, parallel to the ice divide.

## 2.2 The Candidate A site

Van Liefferinge and Pattyn (2013) developed an ensemble model for predicting regions of frozen bed using a combination of remote sensing and teleseismic estimates for geothermal heat flow combined with a thermomechanical ice sheet model calibrated by observations of subglacial lakes. When thresholds for ice thickness (>2000 m) and of the horizontal component of the ice velocity (< 2 m/yr) were applied, a map of possible old ice candidates was produced (Fig. 1).

In the Dome C region, five candidate sites exist, which we term A, B, C, D, and E (Fig. 2). Notably, none of these sites overlaps with the EPICA Dome C ice core site near Concordia Station – consistent with the likely basal melting there implied by extrapolation of borehole temperatures. Sites B, C, and D are located on the steep and poorly sampled subglacial peaks on the northeastern side of the Concordia Subglacial Trench; basal ice in this region has likely traversed the deep, wet Concordia Subglacial Trench. Site E lies on a small subglacial high downstream on the Totten Glacier side of the Dome; this site also lies down flow of a deep subglacial trough, thus raising substantial doubt to its suitability as an old ice coring site.

Candidate A is by far the largest site in the Dome C area and lies under the ice divide on a subglacial massif, minimizing both ice thickness and ice velocity. The ice surface above Candidate A forms a topographic extension to the south of Dome C informally termed 'Little Dome C'. The central part of Candidate A lies 40 km southwest from Concordia Station. The size of Candidate A compared to the 5 km model cell size also makes it more likely that the Van Liefferinge and Pattyn (2013) model captured basal temperatures correctly. Because of its characteristics, Candidate A represents a near term primary goal of European and Australian old ice site selection.

The 2011 airborne survey line (VCD/JKB2g/DVD01a; location shown in green on Fig. 2) crossed the middle of the Candidate A site. Focusing of the radar data showed that the southern flank of the Candidate A massif ended in a steep cliff over which englacial layers dive (Fig. 3). Coherent, continuous englacial reflectors are present in the upper 80% of the ice column (Cavitte et al., 2016), while in the bottom 500 m, a region of more diffuse englacial scattering is present. This distinct zone of basal ice is also apparent in Operation IceBridge (OIB) radar data that operates at a higher frequency (Cavitte et al., 2016; Leuschen and Allen, 2011a), and in appeerence is similar to the 'valley wall' accretion ice seen in Dome A (Bell et al., 2011).

## 2.3 The need for new data

Uncertainties in the older datasets, a lack of resolution appropriate for the small scale processes near the base of the ice sheet, and inadequate knowledge of subglacial hydrology and geothermal heat flow drive a need for greatly increasing resolution over these old ice targets.

### 2.3.1 Poorly positioned data

Figure 2 demonstrates the requirement for additional new data. Surface slopes from high resolution digital elevation models (DEMs) (for example Helm et al., 2014) often correlate with structure in the subsurface (e.g. Ross et al. (2014); Jamieson et al. (2016)); however, in the Dome C region, we see regions of Bedmap2 (outlined in yellow) in which structural bedrock trends significantly disagree with those inferred from ice surface slopes. These regions are either poorly sampled (right yellow box) or, of more concern, only constrained by poorly positioned, pre-GPS SPRI/NSF/TUD radar sounding (left yellow box, over the Concordia Subglacial Trench). Positioning quality for these older sounding data has been reported to be $\sim 5$km, however a 15 km offset along-track would be required to reconcile the surface slope structure and Bedmap2 bed elevation data at this location (as the flight line crosses the trough, the interpolated topography is not sensitive here to cross track errors on this line). The SPRI/NSF/TUD data, along with Soviet data with similar positioning issues are especially problematic for Bedmap2 in the deep interior of the ice sheet where most old ice candidates are found.

### 2.3.2 Small scale relief

Initial radargrams such as that shown in Fig. 3 (upper) show considerable small scale bed roughness, not captured by Bedmap2. As Bedmap2 was designed for a continent wide interpolation of the data, reproducing the small scale variability of the bed was not a priority (Fretwell et al., 2013). However, correct positioning of old ice coring efforts will be highly sensitive to small scale structure (Tison et al., 2015). Radar data that takes advantage of the additional resolution possible through Doppler focusing is essential for understanding the along-track small scale structure of these mountainous regions, while close lines spacing is important for constraining cross-track variability.

### 2.3.3 Subglacial lakes

Subglacial lakes identified from radar are a key constraint on models of basal heat flux (Pattyn, 2010) and are employed by Van Liefferinge and Pattyn (2013) in their model of basal frozen ice. The identification of subglacial lakes is complicated by variations in englacial attenuation that modifies the strong radar reflection due to an ice-water interface (Matsuoka, 2011; Carter et al., 2007), however, new methods independent of radar echo strength that examine the scattering properties of the bed (Schroeder et al., 2015) allow for rapid identification of these 'radar' lakes (Young et al., 2016) in focused phase coherent data (Peters et al., 2007).

## 2.4 The Old Ice A (OIA) survey

Key objectives of the survey were to define the ice thickness at high resolution, infer basal roughness across the target region and map the distribution of subglacial water. Improving the englacial stratigraphy (especially deep layers; Cavitte et al., 2017) and correlating it to the existing EPICA Dome C core site were also high priorities. In addition to the radar data, we acquired laser altimetry, gravity and magnetics data, along with complementary Global Positioning System (GPS) and Inertial Measurement Unit (IMU) data. Instruments are detailed in Table 1.

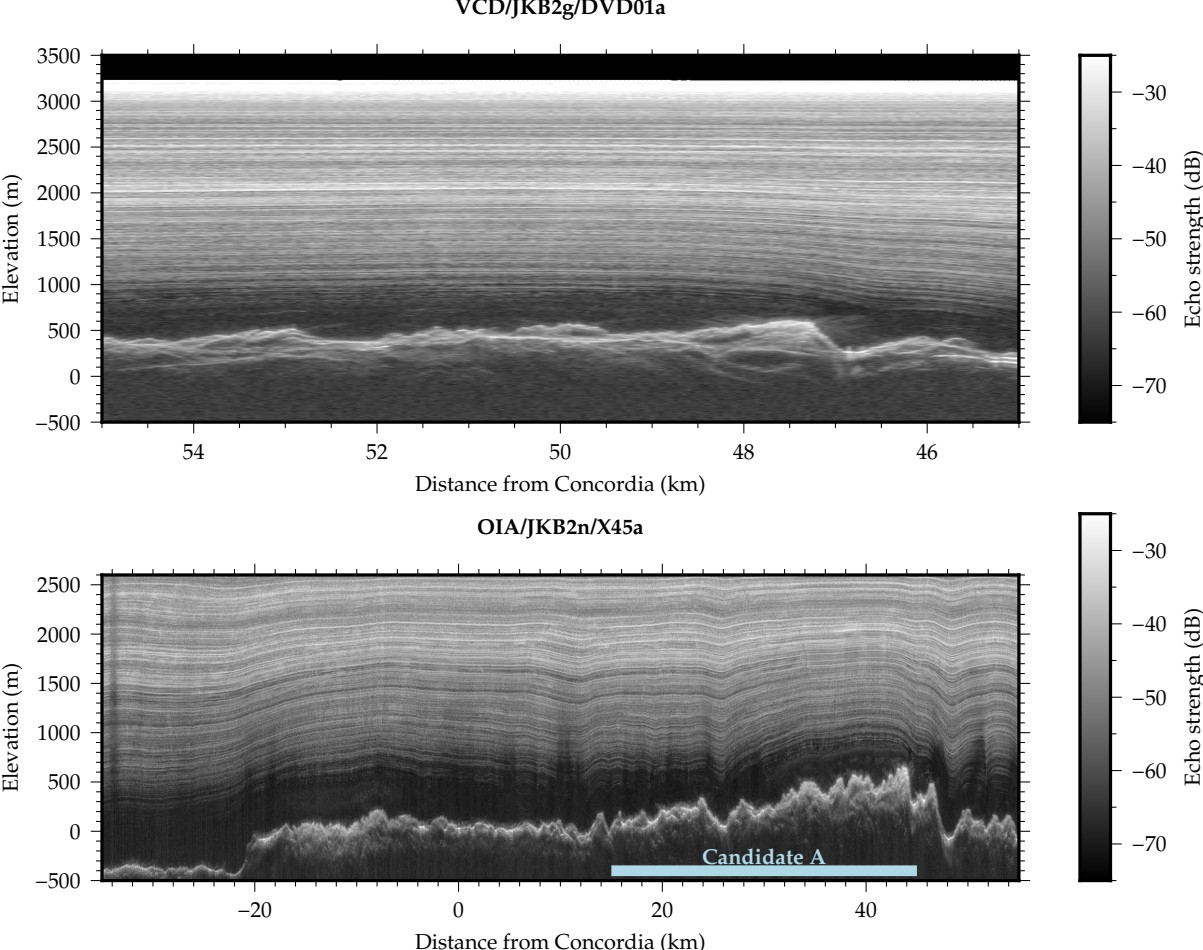

**Figure 3.** top) HiCARS2 2D focused and depth corrected radargram along the ice divide across the Candidate A target; (line VCD/JKB2g/DVD01a from Blankenship et al. (2014) and shown in green on Figure 2). Geographic south is to the right, Dome C and geographic north is to the left, color scale is relative power (geometrically corrected) in dB. Near surface layers have superposed surface scattering. No vertical exaggeration. Echoes appearing below the bed are actually coming from up to a km to each side of the track. bottom) MARFA line from OIA survey (line OIA/JKB2n/X45a) parallel to VCD/JKB2g/DVD01a, showing distance from Concordia, the location of Candidate A and the EPICA/DMC site. Vertical exggeration is 25x, and orientation is the same as above.

## 2.5 Follow-up ground campaign

The results of this survey have been used for follow-up high resolution ground work, using the BAS DOLORES ground radar (King et al., 2009) for further bedrock mapping, ApRES phase tracking radar (Lok et al., 2015) to track vertical strain rates, and the BAS Rapid Access Isotope Drill (UK-RAID, Triest et al., 2014) to acquire thermal gradients in the upper ice sheet, for geothermal heat flow inversions. A location just north-east of the ice divide in central Candidate A ($122^o$ 12' E, $75^o$ 18' S)

**Table 1.** ICECAP (OIA) instrument suite

| Measurement | Instrument | Date (UTC) | Reference |
|---|---|---|---|
| Ice thickness | MARFA Coherent Ice Penetrating Radar | Jan. 24,28,29, 2016 | Young et al. (2016) |
| Ice surface range | Riegl LD90 Laser Distance Meter | Jan. 24,28,29, 2016 | Young et al. (2015) |
| Ice surface range | Sigma Space ALAMO Photon Counting Lidar | Jan. 24,28,29, 2016 | Young et al. (2015) |
| Magnetics field | Geometrics G823A Scalar Magnetometer | Jan. 24,28,29, 2016 | Aitken et al. (2014) |
| Gravity Field | CMG GT-2A Airborne Gravity Meter | Jan. 28,29, 2016 | Greenbaum et al. (2015) |
| Position | Javad Delta 4-Antenna GPS | Jan. 28,29, 2016; partial on Jan. 24 | Greenbaum et al. (2015) |
| Orientation | Novatel SPAN Integrated IMU/GPS | Jan. 28,29, 2016; partial on Jan. 24 | Young et al. (2015) |

was selected for further investigation. If the ground investigation proves fruitful, the SUBGLACIOR drilling probe (Alemany
et al., 2014) will be deployed to measure the in-situ oxygen isotope record, as a pathfinder for a full eventual ice core retrieval.

## 3  Methods

### 3.1  Survey design

The OIA survey was designed to sample Candidate A at high resolution, with 110 km long longitudinal-to-slope 'Y' survey
lines at separations of down to 1 km cutting across the ice divide, and $\sim$ 65 km long transverse-to-slope 'X' tie lines with
separations of 5 km parallel to the ice divide (Fig. 2d). Some of the X lines extend true northeast to cross the Concordia
Subglacial Trench and candidates B, C, and D, while the Y lines extend far enough to the true northwest to cover Candidate E.
   Two lines were added to cut obliquely across the grid: one that tracked over the EPICA Dome C site in order to connect the
ice chronology to the grid and a second line to better constrain an oblique topographic ridge crossing the divide. Flight lines
were designed to avoid Concordia's clean air sector to the south of the station, as well as to allow the aircraft to make VHF
communications with the station before landing. Typical aircraft speeds were 85 m sec$^{-1}$, and flights at altitude were typically
4 hours in duration.

### 3.2  Survey implementation

Four flights of four hours each were carried out from Concordia Station in late January 2016 - the first two (ICP7/F11 and
ICP7/F12) focused on 2 km line spacing Y lines over Candidate A, followed by one flight (ICP7/F13) targeting X lines
extending past Concordia to Candidate B, and lastly one flight (ICP7/F14) focused on increasing the line density over the
primary target to 1 km line spacing. Initial interpretation of the radar data was performed during the field program, and helped
to refine the later flight plans. GPS base station data were collected during the survey flights.

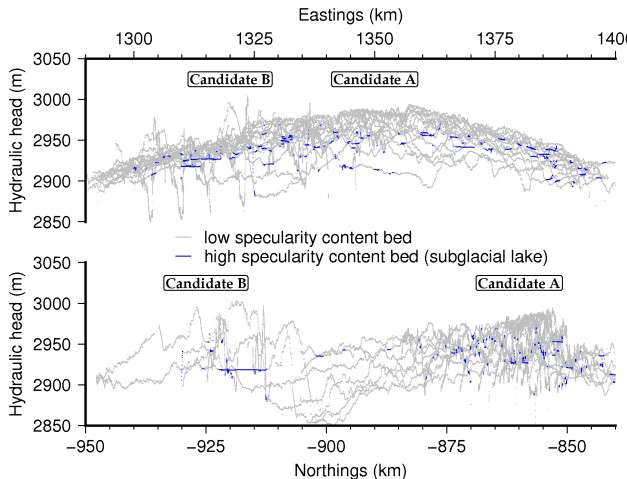

**Figure 4.** The OIA extracted bed data projected into hydraulic head (the water level equivalent to the pressure imposed by the ice overburden), and viewed in the projected Northing plane (lower, looking across the ice divide) and projected Easting plane (upper, looking along the ice divide). Regions of high specularity content (subglacial lakes) are highlighted in blue. Note that lakes are flat, indicating hydrostatic equilibrium. Many of the subglacial lakes lie in valleys that cut into the primary surface that envelopes the local subglacial topography; only one is found above the 2950 m head level. Projection for the horizontal axis Eastings and Northings is Antarctic Polar Stereographic.

### 3.3 GPS and laser altimetry processing

After the field season, GPS data were processed using Waypoint Inertial Explorer, using Precise Point Positioning (PPP) loosely
coupled to the acceleration and roll rate data from the SPAN IMU system. Internal estimates of uncertainty for these data have 2 cm vertical standard deviation, and 4 cm horizontal standard deviation. Apparent surface elevation differences between survey lines at crossovers were minimized to obtain laser altimeter pointing biases (using the methods described in Young et al., 2015).

### 3.4 Radar processing

Radar data was stacked in acquisition 32 times and were coherently recorded at 196 Hz; these data were range-compressed
resulting in a range resolution in ice of 8.4 meters (Cavitte et al., 2016). The radar data was first processed using a very short synthetic aperture (Holt et al., 2006; Young et al., 2011) to extract the surface return and for initial quality control. This processing (called "pik1") retains the unmigrated along-track hyperbolae that characterize many earlier radar sounding datasets. The data was then processed using the "1-D" focused SAR approach of Peters et al. (2007), where focusing of the along-track Doppler phase variations within each range resolution cell was employed to improve the along-track resolution to
approximately 10-20 meters for scattering targets. The data was resampled to 4 Hz along-track sampling (~22 m along-track sampling), and the logarithm of signal power was displayed for manual interpretation.

### 3.5 Radar ice thickness and bed elevation extraction

To obtain ice thicknesses, we systematically select a window around the earliest bed return, and then automatically select the best fitting pulse waveform within that window (assumed to be a paraboloid power profile in decibels), for both the surface and the bed. The surface time delay is subtracted from the bed time delay to obtain the two way travel time in the ice column and, using an appropriate refractive index for ice ($\sqrt{3.15}$), we convert to ice thickness. We choose not to apply a firn correction to ice thicknesses; as shown in (Peters et al., 2007), a firn correction is not required for our focusing, and and will not affect the conclusions in this paper (firn correction is however critical for isochrone interpretation Cavitte et al., 2016; Winter et al., 2016). Bed elevations are derived by subtracting the ice thickness from concurrently collected laser or radar altimetry; all elevations are referenced to the WGS-84 ellipsoid.

We do not attempt to reconcile ice thickness interpretations at crossover points, and maintain a strict first return policy. The first return represents a stable interface to interpret in radar, but has a high likelihood of selecting off nadir echoes in steep topography. As detailed in Appendix A, preserving crossover differences provide important information on understanding the interactions between radar geometry, processing, and bedrock roughness, and allows us to extrapolate these statistics to intervals without crossover constraints.

### 3.6 Subglacial lake detection

Specularity content of the basal return was extracted by comparing the echo strengths of the bed from "1-D" focused SAR to the results of range-migrated 2-D focused SAR, following the approach outlined in Schroeder et al. (2015). Regions with a specularity content of greater than 0.2 were classified as subglacial lakes.

Hydrostatic pressure is important for the context for subglacial lakes, and is often represented by hydraulic head (equivalent to the height of a water column with the same basal pressure as the ice load). Gradients in hydraulic head control basal water flow direction; the magnitude of the slope of hydraulic head controls the expression of water flow. In this case, hydrostatic equilibrium, indicated by zero hydraulic head gradient, was not used for subglacial lake identification (as was the case for the 'lake detector' in Carter et al. (2007)), however all subglacial lakes that were identified had low hydrostatic gradients (Fig. 4).

### 3.7 Ice thickness compilation

We combined the OIA results processed as described above with older datasets from Italy (Tabacco et al., 1998), Germany's Alfred Wenger Institute (AWI) (Steinhage et al., 2001), Operation IceBridge (OIB) data (Leuschen and Allen, 2011b), and British Antarctic Survey (BAS) data from (Jordan et al., 2010) (of these datasets, much of the Italian data and AWI data had been included in Bedmap2). We exclude the poorly positioned SPRI/NSF/TUD data.

We compared these surveys to a 1-km resolution grid derived from OIA focused bed elevation data, and found good visual matches between the coherent, focused BAS and OIB data (see Table 2). The negative bias in OIB data is likely due to the resolution of small scale valleys in the OIB profile data that are not resolved in the 1 km grid we used for this comparison.

**Table 2.** Bed elevation dataset comparison to an OIA focused data grid, with mean offset and standard deviation $\sigma$ of bed elevation difference.

| Provider | Processing | Mean* | $\sigma$ |
|---|---|---|---|
| BAS (Jordan et al., 2010) | focused | -3.4 m | 60 m |
| OIB (Leuschen and Allen, 2011b) | focused | -21.2 m | 42 m |
| Italy (Tabacco et al., 1998) | unfocused | -35.4 m | 44 m |
| AWI (Steinhage et al., 2001) | unfocused | 46.6 m | 55 m |
| OIA full resolution data | focused | -0.4 m | 27 m |

\* Bed elevation difference; positive is higher than the OIA-only grid

**Table 3.** Bed elevation dataset comparison to Bedmap2**, with mean offset and standard deviation $\sigma$ of bed elevation difference

| Provider | Bedmap2 inclusion | Mean* | $\sigma$ |
|---|---|---|---|
| BAS | absent | 20 m | 110 m |
| OIB | absent | 34 m | 77 m |
| Italy | partial | 63 m | 62 m |
| AWI | included | -5.6 m | 77 m |
| OIA full res. | absent | 9 m | 133 m |

\* Bed elevation difference; positive is higher than Bedmap2

\*\* All data was converted to the WGS84 datum

Both the Italian and AWI datasets were acquired incoherently without focusing or migration, which will induce range hyperbolae in the radargram that will tend to reduce the measured ice thickness. This does not appear to have been a major effect on the biases, however. For the Italian data, the lower mean value is due to the coarse resolution of the pulse, combined with noise in the picker used for the Tabacco et al. (1998) survey; for incorporation into the compilation, we select the shallowest return in each 1 km block of data. For AWI, ice thickness measurements at peaks are systematically $\sim$50 meters smaller than for OIA. This corresponds to the length of the high energy pulse used for deep ice sounding. We add 50 meters to the AWI ice thicknesses for incorporation into this grid. In all cases, the standard deviation of the trackline data compared to the OIA-only grid was better than the comparison to Bedmap2 (see Table 3), likely related to the loss of spatial resolution in Bedmap2.

A compiled grid using all of these data was generated by first extracting the median value for the data at 500 m cells, and used a natural neighbor interpolator (nnbathy, Sakov (2016)) on the data. We apply a 2 km Gaussian filter, and mask out data more than 5 km from a datapoint.

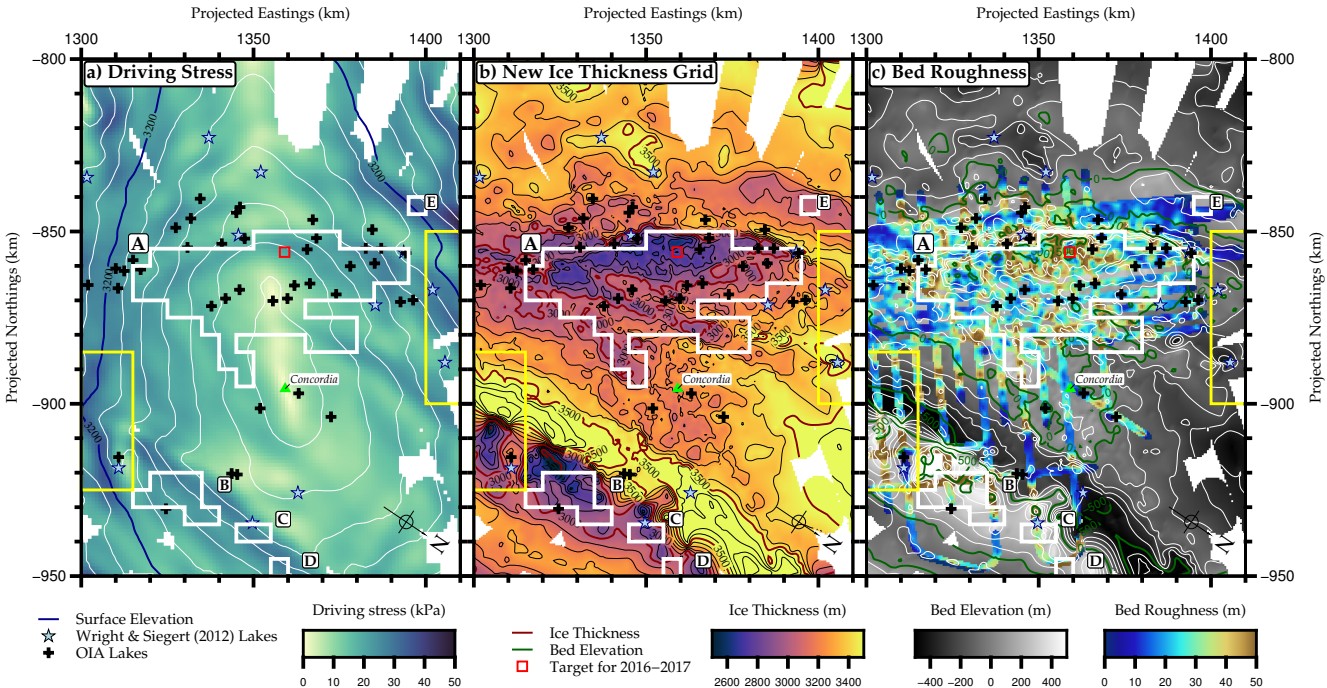

**Figure 5.** The Dome C old ice candidates in the context of updated datasets. New subglacial lakes, identified using specularity content, are shown as black crosses; regions of surface-bed disagreement in Bedmap2 are shown in yellow. a) Driving stresses using the new ice thickness data and the Cryosat-2 surface DEM (Helm et al., 2014) (smoothed by 15 km to remove longitudinal stress gradients) show that all of the candidates aside from the innermost portion of Candidate A lie over regions of relatively high (20-30 kPa) driving stress (note that surface slope is the primary driver on driving stress here). b) Compiled ice thickness data provided by ICECAP, AWI, BAS and INGV. c) RMS deviation of the bed at 800 m length scale using OIA data only, superimposed on new bed elevations (in grayscale). The region tends to be rougher toward the center of the Candidate A region, and smoother toward the edges and in the troughs. Concordia Station (green triangle) lies in a particularly smooth area. Projection is Antarctic Polar Stereographic.

## 4 Results

We use the new and compiled sounding data to evaluate the roughness of the interface and the subglacial hydrological context for the region and to investigate the stress state of the ice.

### 4.1 Ice thickness and bedrock topography

While the outlines of the terrain at the 10 km length scale were visible in Bedmap2 (Fig. 2 a), which was largely derived from the Tabacco et al. (1998) survey (Fig. 2 b), the addition of the OIA dataset delineates the key features of this landscape (Fig. 5 b,c). The Concordia Subglacial Trench is bound by a sharp, west-facing dissected escarpment approximately 2000 meters relief that hosts Candidate B, C and D. In this new compilation, with the pre-GPS data removed, this escarpment is

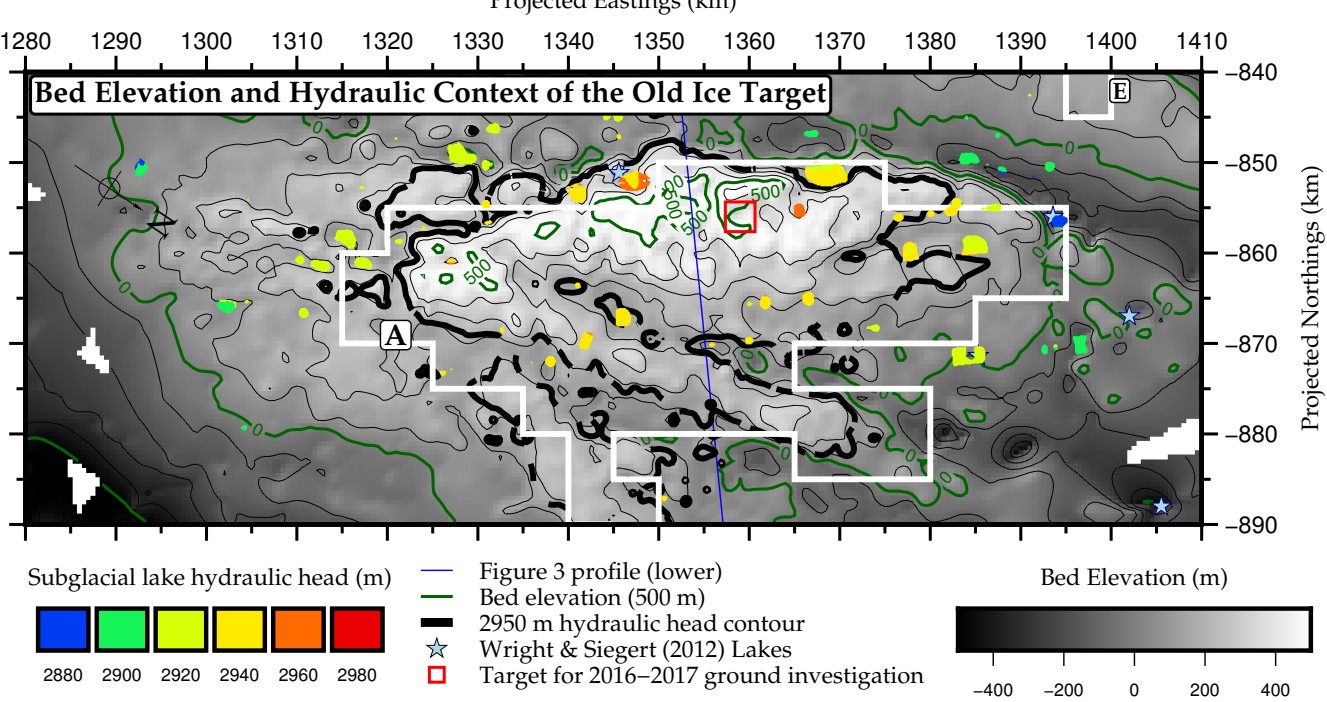

**Figure 6.** The selected target within Candidate A. Compilation bed elevation with imaged subglacial lakes superposed. Target region for further ground work lies near the highest point on the bed, northeast of the ice divide.

no longer discordant with the surface data (left yellow box in Fig. 5). Instead we see an array of hanging valleys, consistent with a glaciated terrain. The minimum ice thickness of 2383 meters in this region occurs on a sharp peak between two of these hanging valleys.

The massif underlying Candidate A is bound by a southwest facing 200-300 meter high system of scarps to the southwest, which capture a system of perched lakes. The massif dips gently to the northeast, and is marked by a series of 200 m deep, 2-3 km wide valleys running toward the north, divided by occasionally large ridges. Under the divide, there is a local 700 m elevation peak where ice thickness is under 3000 m. The minimum ice thickness within Candidate A (2600 m) occurs on a local peak adjacent to the southern escarpment.

To the southeast, a complex series of troughs with extensive water bodies emerges from the Candidate A massif and opens out into the Concordia Subglacial Trench.

### 4.2 Additional subglacial lakes

Using specularity content, we map out 54 subglacial lakes in the OIA survey, 50 of which were not included in the Wright and Siegert (2012) compilation. Details on these lakes are provided in Supplementary Materials. The largest of these lakes is 11.5 km long and lies in a hanging valley on the northeast side of the Concordia Subglacial Trench (located at 1310 km,-915 km).

A second large lake, at least 4.5 km long, lies within the Candidate A region, in the escarpment to the southwest of the massif that underlies "Little Dome C" (located at 1367 km, -852 km). 50% of segments of specular bed that were 1 km or greater in length had hydraulic head gradients less than 0.1%, meeting the criteria for a lake in Carter et al. (2007), and 71% were less than 0.2%. This result is consistent with flexural support of small gradients around the edges of these small lakes (Carter et al., 2007). In all, 19 lakes are now observed in the region predicted to be at least as cold as -5 C, only one of which was known to Van Liefferinge and Pattyn (2013).

### 4.3 Small scale roughness

Small scale roughness, at length scales of the line spacing and below, is relevant for four reasons: 1) roughness gives insight into the pathways that basal ice must traverse; 2) roughness may provide information on past ice sheet behavior and basal conditions, 3) roughness is a key control on the uncertainties inherent in profiling radar systems, and 4) basal roughness forces the overriding ice sheet to develop a complex deformation pattern in the lower part of the ice column, and this deformation field could disturb stratigraphic continuity of the ice core record.

We calculate the along-track roughness as the deviation in detrended elevation between points of a given length scale (Shepard et al., 2001; Young et al., 2011). For a given cell size, we calculate the Root Mean Square (RMS) deviation. We choose 800 meters length scale for this investigation, as it both provides insight into sub-line spacing roughness, but is also relevant for the cross-track beam pattern for understanding uncertainties in the ice thickness data (see Appendix A).

Typical RMS deviations at 800 meter length scale are 40 to 50 meters toward the center of Candidate A, and are lower toward the margins (Fig. 5 c), although locally smoother regions 3-5 km across exist in places. One of these locations is the EPICA Dome C ice core site. Much of the base of the Concordia Subglacial Trench, and the regions surrounding the massif, are very smooth, consistent with deformable sediments.

On the massif, smoother regions also tend to correlate with regions of subglacial lakes, although subglacial lakes are also found in extremely rough regions with deep incisions (for example location 1320 km, -860 km on Fig. 6). Much of the regions of higher roughness in central Candidate A are sinuous, and appear to follow local valleys, with smoother regions between.

### 4.4 Surface DEM validation from laser altimetry

We used the OIA laser altimetry to validate available satellite based DEMs, using the WGS84 datum. We compared with both the Bedmap2 surface DEM (Fretwell et al., 2013), which in the interior is largely based on the combined ICESat/ERS radar altimetry product of Bamber et al. (2009), and the Cryosat-2 DEM (Helm et al., 2014), wholly derived from radar altimetry. We transformed all DEM's into WGS-84, and used GMT's grdtrack (Wessel and Smith, 1998) to extract points of comparison for each laser point (we removed one anomalous point over Concordia Station itself). Results are shown in Fig. 7.

We find that both DEMs have a significant bias, outside the previously demonstrated accuracy of the ICECAP laser system, with the laser altimetry at Dome C (consistent with some penetration by the radar altimeters) however both have very low levels of noise in this very flat region. There is a slight preference for the Cryosat-2 DEM, which is what we used as our reference ice surface for this paper and for calculating the driving stresses in the following section (e.g. Fig. 5 c).

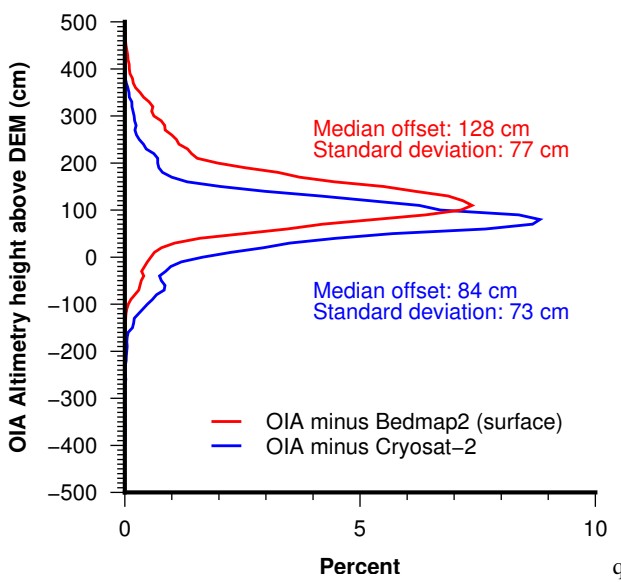

**Figure 7.** Histograms comparing 2 surface elevation DEMs (largely derived from radar altimetry) and OIA altimetry. Expected accuracy of OIA altimetry is 15 cm (Young et al., 2015).

### 4.5 Stresses in the Dome C region

Disturbed ice at the basal interface will be more likely if the ice is under horizontal gravitational stress due to surface slopes; however, decreased horizontal stresses may also be due to basal melting, which would also destroy the sought climate record. We calculate the driving stress to investigate this potential impact on old ice.

We derived surface slopes from a smoothed version (15 km gaussian filter) of the Cryosat-2 DEM, and combine this with the new ice thickness compilation to derive driving stress ($\tau$, Fig. 5 a) using the following formula:

$\tau = \rho_{ice}gh\sin(\theta)$

where $\rho_{ice}$ is 910 kg m$^{-3}$, $g$ is the acceleration due to gravity of 9.8 m s$^{-2}$, $h$ is the ice thickness and $\theta$ is the slope of the ice sheet surface. Surface slopes (compare with Fig. 2 c) dominate the driving stress map.

## 5 Discussion

The results of these data have implications for locating old ice. We discuss them in the context of the five requirements laid out in the introduction

### 5.1 Accumulation history

The englacial reflectors imaged as part of this survey represent isochrones that can be dated at the existing EPICA Dome
C site using the methods outlined in Cavitte et al. (2016), and inverted for basal age. This work is now in progress, with

the stratigraphy developed by Cavitte et al. (2016) being propagated through the entire OIA survey, and are the subjects of follow-up papers (Parrenin et al., 2017; Cavitte et al., 2017).

## 5.2 Geothermal heat flow implications

Van Liefferinge and Pattyn (2013) used the presence of subglacial lakes to calibrate their geothermal heat flow model, and the significant amount of free water in the Candidate A area may cause some doubt as to the prediction of basal freezing. However, it is clear that most of these lakes lie within local valleys in the bed rock. Most lakes do not lie under ice less than 3000 meters thick (Fig. 5b), and Fig. 4 demonstrates that the majority of lakes lie below the enveloping surface of the massif.

When projected in terms of hydraulic head (the height of water consistent with the overburden pressure at the bed), few lakes appear above 2950 m head, implying the presence of a limiting hydraulic or thermal 'water table' (Fig. 6). The implication for these lakes on regional geothermal heat flow may be limited by local topographic focusing, which may locally double basal heat flow (van der Veen et al., 2007). This factor is not taken into account in the Van Liefferinge and Pattyn (2013) model due to its 5 km spatial resolution, but may be significant in the interpretation of subglacial lakes in this deeply incised region. Notably, there should be conservation of energy; geothermal heat flow will be reduced in the regions between valleys, and latent heat absorbed by melting ice in the valleys will not be available for melting on the highs between valleys.

The primary target for ground work is 10 km upstream from the nearest subglacial lake at a similar hydraulic level (Fig. 6), futher indicating the need for high resolution work in this area.

## 5.3 Glaciological context

Only Candidate A lies over the divide. The other frozen bed candidates lie well off the divide, and lie down stream of significant subglacial topography. In addition, observed driving stresses (derived using a validated surface elevation DEM), provide additional context, although the interpretation is not straight-forward.

Without reliable velocity data, a formal inversion for basal shear stress is not useful, however, driving stress and basal shear stress are often well correlated (Sergienko et al., 2014). We see in Fig. 5 elevated driving stresses correlated with thinner ice (Fig. 5 b) and elevated bed topography (Fig. 5 c). The relationship with bed structure implies that this configuration of driving stresses have not evolved much over time in response to divide migration.

We see elevated driving stresses over Candidates B, C, D and E, as these candidates lie on local mountains, and over the southern edge of the Candidate A massif. Basal ice in these locations may be disturbed by the application of these elevated driving stresses. Within Candidate A, regions with low driving stress on either side of the ice divide have a population of subglacial lakes, implying an element of basal lubrication, and thus melting may have modified the base of the ice. In the upland region on the divide, driving stresses increase over the southern escarpment. The targeted region (Fig. 5 a) has intermediate to high driving stresses, that may be an unavoidable consequence of targeting a shallow area where the ice surface is at its highest.

## 5.4 Basal roughness

While the results of Tison et al. (2015) indicate that basal ice may be sensitive to elevated basal roughness, conversely low basal roughness may also indicate conditions not favorable to the preservation of coherent basal ice. We observe a spatial relationship between subglacial lakes and larger smoother areas, including near the EPICA Dome C ice core site (Fig. 5).

Areas of elevated basal roughness appear to be associated with valleys in the topography, and again subglacial water. However, there does appear to be regions between the incised, rough valleys with reduced roughness, in the interior of Candidate A on the order of 5 km across.

## 5.5 Ice thickness

In the Dome C region, we found no ice thinner than 2500 meters that lay over subglacial topography that was appropriately flat. However, we found a significant region with overlying ice thinner than 3000 meters that was lacking in free water bodies, and contained kilometer-scale flat regions. This was consistent with the coldest region of the Van Liefferinge and Pattyn (2013) model.

## 5.6 Prospects for old ice at Dome C

The Dome C region, as mapped by the OIA survey, challenges intuition regarding old ice targets. Subglacial lakes are common, the bed is rugged in key places, and the ice is not as thin as recommended (Fischer et al., 2013), although this general guideline was to help avoid melt water, which is here identified by specific means. However, a detailed inspection of the data is encouraging.

The best compromise target is the center of the massif, near the ice divide of Little Dome C, in the Candidate A region (Fig. 6). Flatter regions lie between incised, rough valleys that serve to capture geothermal heat flow and melt water; and thus the rough topography of the Candidate A region may serve to preserve old ice on elevated areas between the valleys.

However, a trade off is that maintaining a simple flow path for basal ice in such an rough environment will be difficult, and the mountainous region also induces relatively large driving stresses in the overlying ice. The paths taken by basal ice elements in such an environment may be torturous, and result in stratigraphic complexity. These compromises may be a requirement of finding the necessary 'flat mountain' for 1.5 million year old ice. Detailed site selection work (currently in progress), and careful, 3D modeling of geothermal heat flow in the context of rough topography will be required.

## 6 Conclusions

An international program conducted a successful high resolution, multi instrument survey of a key old ice region in the Dome C region of East Antarctica. We found that Candidates B, C, D and E either lie on extremely steep and rough topography, or lie downstream of deep, smooth troughs, implying transport and melt may have compromised the old ice record.

Candidate A has some promising sites, including a shallow peak directly under the divide; however, a large number of subglacial lakes and generally rough terrain, present challenges to site interpretation and selection. Ongoing modeling of this data (including englacial structure) and high resolution surveying are in progress to evaluate these targets.

## Appendix A: Quantifying uncertainty in focused radar data

Crossover differences in ice thickness (or equivalently bed elevation) between radar lines are often reported as a metric of uncertainty in the quality of the ice thickness data. However, given the geometry and processing of radar sounding data, the information contained in these crossovers must be carefully considered. As well as the inherent science interest in the uncertainties in the data, the density of orthogonal lines over thick ice and a rough bed target presents an opportunity to better understand the nature of uncertainties in this kind of dataset in general.

Crossover bed elevation statistics were computed using just the orthogonal X and Y lines of the OIA survey. For focused (foc1) data, the RMS difference is 80 m; for unfocused (pik1) data, it is 54 m. The result is counterintuitive; the more intensive processing has higher crossover differences. This difference can be explained by understanding the geometric controls on the radar signal and the interactions with bedrock roughness.

### A1 Incoherent sounding crossover differences

In the case of the incoherent pik1 processing, the beam pattern is effectively limited by the critical angle of refraction of the air-ice interface (34°), in both the along-track and across-track directions. The processed radargram in this case effectively shows the range to the nearest bed interface, and the direction of travel does not affect that range.

Approaching the crossover point from either direction, a similar range is seen, even if the reflecting target is not under the aircraft. If the first return is coming from a range-compressed target, an incorrect (and likely too thin) ice thickness will be inferred; this is an error that will not be indicated by the crossover difference. In general, in rough terrain, unfocused data will provide a considerable underestimate of ice thickness of up to tens to hundreds of meters in valleys.

### A2 Focused sounding crossover differences

In focused radar data, discontinuities are often seen in crossovers, especially where terrain is rough or steeply sloping. These discontinuities are due to the asymmetry in resolution between the fine along-track resolution (10-20 meters), determined by the synthetic aperture generated by motion of the radar, and the coarser across-track field of view, determined by the real aperture of the two under-wing dipoles. The across-track beam at the bed covers approximately 1 km either side of the nadir point.

Due to the refraction of ice, the wavefronts propagating to the bed are wide parabolae, meaning that small scale topography projecting above the nadir bed can lay over the nadir return. The result is that the first return will tend toward the minimum ice thickness within the aircraft beam pattern, however the measured thickness at the site of reflection will be slightly overesti-

**Crossovers versus Roughness**

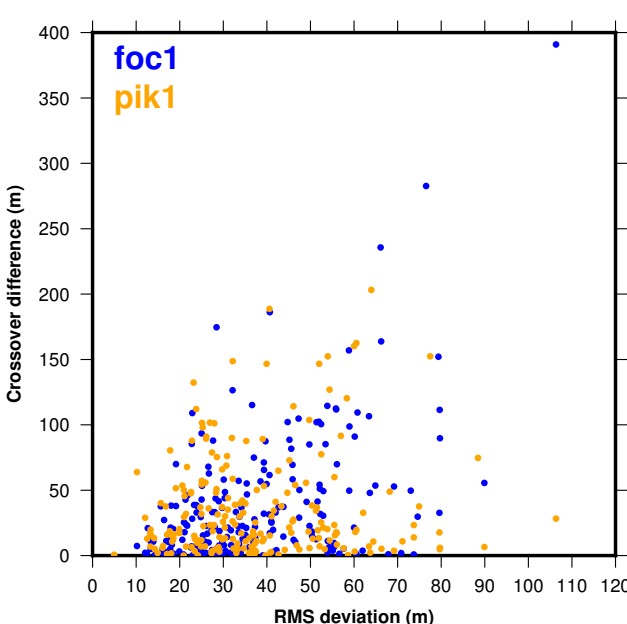

**Figure 8.** Relationship between RMS deviation at 800 m length scale (as measured in the focused bed elevation data) and crossover difference in bed elevation. The focused data (foc1) has larger outliers in rough terrain, as one direction is actually more correct; for the unfocused data (pik1), the crossover is smaller, as both directions are equally wrong

30   mated. The primary uncertainty will be in the cross-track position of the bed echo. Alternatively, if it is assumed that the echo is from nadir, the inferred ice thickness will tend to be underestimated.

### A3   Roughness control on crossover differences

The apparent large scale roughness of a radar profile will be dominated by along-track roughness, but smoothed by layover contributions from the side. Fig. 8 shows the relationship between RMS deviation at 800 m length scale (as measured in the focused bed elevation data) and crossover difference in bed elevation. In both processing approaches, there is a roughness correlation on maximum crossover difference. A stronger relationship is seen for the focused data than for the unfocused data,

5   primarily due to the geometric arguments given earlier. In the case of the unfocused pik1 data, it is a case of both survey lines being equally wrong. Therefore, empirically, the uncertainty in ice thickness for both focused and unfocused data is about three times the observed local along-track roughness at 800 meters.

    The key result of this analysis is that maximum crossover discontinuities may be predicted from along-track roughness measurements, and assuming isotropic landscapes, the spatial variation in ice thickness uncertainty may be inferred from

10   sparse, non-crossing lines. For areas of large roughness values, the horizontal position of the aircraft GPS cannot be assumed

to represent the location of the ice thickness. This knowledge may help guide future data acquisition, as well as how ice sheet models ingest profile data.

*Author contributions.* D. Young wrote the manuscript. D. Young, J. Roberts, C. Ritz, E. Quartini and C. Tozer were involved in the ICECAP II data acquisition at Concordia Station. S. Uribani, D. Steinhage and H. Corr contributed older data. All authors helped conceive the experiment, design the flight plans and edit the manuscript. The authors declare that they have no conflict of interest.

## Appendix B: Data availability

NetCDF3 files of the compiled bed elevation, ice thickness and bed roughness are included in the supplementary materials. ASCII comma delimited tables of the new subglacial lakes, and ice thickness point data in a shared ASCII space-delimited format for each contributor are
5 also included.

*Acknowledgements.* This research was made possible by the joint French–Italian Concordia Program, which established and runs the permanent station Concordia at Dome C. The Australian Antarctic Division provided funding and logistical support (AAS 3103, 4077, 4346). This work was supported by the Australian Government's Cooperative Research Centre's Programme through the Antarctic Climate and Ecosystems Cooperative Research Centre (ACE CRC); support for UTIG came from the G. Unger Vetlesen Foundation and NSF grant PLR-
10 1443690. We acknowledge the support of Kenn Borek Airlines, in particular J. Chistom, J. Gilmore, and A. Dumont. Quantarctica, QGIS and the Generic Mapping Tools were used for both survey design and data analysis. We thank G. Muldoon, G. Ng and A. Jones for assisting in this work. Three anonymous reviewers provided suggestions that greatly improved this work. This paper is UTIG contribution 3087.

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
