# Peer review of "High resolution boundary conditions of an old ice target near Dome C, Antarctica"

_The Cryosphere, 2016_

## Referee Comment (RC1) · Anonymous Referee #1 · 29 Aug 2016

**General comments**

Young et al. presented compilation of ice thickness data in an area around Dome C, combining the data that Young et al. obtained through their ICECAP II project and several other sets of older unpublished data that British, German and Italian parties obtained. The aim of the authors are to provide updated information on boundary conditions (ice thickness and bed roughness) in Dome C area in East Antarctica, where drilling of very old ice cores may be planned in future. Ice thickness data were collected with flight line spacing of 1 km. In addition, crossover differences in ice thicknesses were evaluated. The main body of the data were obtained from a radar with phase analysis, called as SAR processing, removing along-track hyperbolae effects of the radio echoes. The authors provided some discussions on the crossover differences caused effectively by highly elliptical shape of the radar footprint. At the end of the paper, the authors provided a new bed topography map as compilation of their ICECAP II project data and the other older data. The authors also performed evaluation of bedrock roughness in terms of RMS over a horizontal distance of 800 m along the flight lines. The authors suggested that there are some promising sites in a region of "A" for future ice coring.

This manuscript provide new information of ice thickness in this area with technical issues related to processing. My viewpoints are as follows.

(i) Main subject of the manuscript is improvement of information on boundary conditions (ice thickness and bed roughness). There is a large achievement in terms of making maps.

(ii) There are discussions on bed roughness, in terms of the effective shape of the footprint (caused by the SAR processing along the track).
There seems to me some problems and large rooms of improvement. I will explain details of them in the specific comments.

(iii) The ice thickness data with the SAR processing and the authors' procedure of echo extraction (always the first bed return and not the strongest bed return) were compiled with the older data with no information (or evaluation) of data quality. In addition, no evaluation of crossover errors are given between the new ICECAP II data and the old unpublished data. This situation causes question in terms of quality of the new compilation.

(iv) The authors also performed evaluation of bed roughness in terms of RMS over a distance of 800 m. This is a positive point though the analysis still seems preliminary.

(v) The authors conclude the candidate area A has some promising sites for future ice coring within it. But this conclusion does not seem to be based on good scientific discussions.

In the context of sites for drilling very old ice cores, main readers of the manuscript will be ice core scientists. They will want to see items as follows.

(i) Relation between detailed updated bed topography and known distribution of the subglacial water, both lakes and distribution of the wet/dry conditions of the bed. This is something that the authors can show relatively easily using published data and their new sets of data.

(ii) Distribution of the internal layers is also something that the authors can show relatively easily using published data and their new sets of data. The authors could show how dated layers near EDC ice core are traced over the candidate area A and the other candidate area. In addition, the authors

could discuss information of layers near the base of the ice sheet over the candidate area A

(iii) Like earlier BEDMAP and BEDMAP2 papers, I expect that electronic version of DEM data and the other related data should be published as supplementary data of this manuscript. In addition, I expect all sets of ice thickness data used in this work should be published.

Besides, I point out that  directions in figures and descriptions seem highly confusing condition in this manuscript at the moment.

Overall, as a technical report for the boundary conditions, this manuscript has large rooms to repair, to improve and to add more items. Scientifically, more discussions for boundary conditions and suitability for the oldest sites are needed. When the authors publish maps, digital materials of DEM and raw sets of ice thickness data should be published as supplementary information. Orientation of the maps and descriptions in the manuscript should be carefully checked. If necessary, they should be repaired.

**Specific comments**

Abstract
It is too concise with about 100 words. Please use space of the abstract efficiently. Please use much more words and space (2~3 times of present length) to explain what are really new in the manuscript in terms of science. It seems scientific new is almost simple improvement of the topography map.
The third sentence in the abstract is not easily understandable. In addition, the authors did not show distribution of lakes or water in the paper. Then, please do not mention in the abstract. Otherwise, please show distribution of lakes and subglacial water in the manuscript.

P2L6
I did not find candidates A - E in cited papers. Please explain.

P2L9 - 10
Please provide citations for ICECAP and ICECAP2.

Section 2
Please provide a map showing entire Antarctica and Dome C region in it. In addition, please provide a map with site locations such as Totten Glacier, Byrd Glacier or George V Coast, VCD corridor. Such maps will help readers. Readers are not necessary familiar to this Dome C region of Antarctica.

P2L20  "crude" -> "pioneering"? I suggest so. Of course, pioneering work is usually crude.

P2L21 Why "however"? Did somebody question presence of dome?

Figure 1
Caption
"OIA" appeared here without any explanation before.
Background MODIS data has no contrast within it. Thus there is no meaning to show it here.
"Antarctic Polar Stereographic" Please provide standard latitude.

Figure
Please indicate X and Y directions because they are often mentioned in the text. In addition, horizontal axis is used as Y by the authors.

Section 2.1
Please provide much more information of the radar system used for this work, such as peak power, pulse compression rate, antenna gain, beam (half power) width in E direction and H direction, effective thickness resolution in ice. Perhaps it may be found in the Peters paper or the other papers. However, it is not kind at all for the authors not to show concrete information to readers in this paper. What is "focusable data"? Please explain to readers kindly and properly.

P3L14
What is Vostok/Concordia/DDU corridor (VCD)? Please explain using a figure. What is DDU? Please think about readers who are not familiar to this region.

Figure 2
Please indicate area A - E once again in this figure. Otherwise, readability is not good.
Please indicate flight lines of Figure 1 once again on this map to make better readability.

P4L4-5
*Sites B, C, and D are located on the steep and poorly sampled peaks on the northern side of the Concordia Subglacial Trench (CST)*;

Does it mean bed or surface?
B, C and D seems on the southern side of the CST in Figure 2. Am I wrong? Then why?
The authors seem to use too much symbols such as DDU, CST, VCD and so on. It seems too much for readers. Later in the manuscript, I felt hard to remember their meanings.
Where is CST on the map? It is hard for readers who are not specialist for this region.

P4L6
*basal ice in this region likely traverses the deep, wet CST and is unlikely to be stratigraphically intact.*
It seems still a vague guess. Mountainous area has at least width of ~10 km. Can you suggest some direct/indirect evidence, for example, internal layers?

P4L11-12
*The ice surface above Candidate A forms a topographic extension to the south of Dome C informally termed 'Little Dome C'. The central part of Candidate A lies 40 km south from Concordia Station.*
It seems south and north are very confusing in this manuscript and maps, like X and Y. Please make them very clear to readers.

P4L13
*VCD/JKB2g/DVD01a* is a kind of jargon for readers.

P4L13-14
*Focusing of the radar data showed that the southern flank of the Candidate A massif ended in a steep cliff over which englacial layers dive.*

First, I was confused in terms of directions.
Second, "dive" seems inadequate because the authors used very high vertical exaggeration of ~20 times in Figure 3. If the authors use real scale, it should be very smooth, flat and continuous layers. "dive" is just an artifact effect that the authors produced by exaggeration.

Figure 3
Did the authors apply the geometrical spreading effect in dB? Or, are these data just return power from targets? Please make this point clear for readers.
Please indicate south and north. Please indicate this segment of the flight line on the maps (Figures 1 and 2).
This figure is given but not discussed in the manuscript.

Figure 3
I suggest that roughness and amplitude/frequency of it should be analyzed using such data here. It is much better data source compared to the crossover differences or RMS that the authors are discussing in this manuscript. For example, at a site of X=50 - 55 (km), difference between the first echo and the strongest echo is as large as 200 m or more. By choosing only the first echo for ice thickness analysis, the authors ignore the strongest echo which is most probable echo from the nadir. With this reason, the authors analytical produce is causing a bias of underestimation of ice thickness

from the beginning. In addition, analysis of the both leading and trailing edges will give very good measures of the bed roughness.

P5L7 What is Internal Measurement Unit? Please explain to readers.

Title of the section 3
Explanation of the OIA is given only later. Please explain to readers.

Table 3
This seems a strange table to me with several reasons.
(i) Why commercial names of the instruments appear in the first column? Item of measurements should be given first such as ice thickness, distance between aircraft and the ice sheet, position, etc.
(ii) F11 - F14 are not given well in this manuscript. It is hard to understand. Only the authors know well.
(iii) Please give priority to instruments that you really used for discussions of this manuscript. Gravity and Geomagnetic are not discussed in this paper. Radar sounder and positioning should be shown with higher priority here.
(iv) What is ICECAP2? What is different from ICECAP? Few readers know them.

P6L4-5
*Flight lines were designed to avoid Concordia's clean air sector to the south of the station, as well as to allow the aircraft to make VHF communications with the station before landing.*

I suggest this should be removed because it is something that only very limited people should know.

P6L14 "Elevation difference"
Does it mean elevation of the ice sheet surface? Then, please clarity it to readers.

Section 3.2
Please prepare a figure showing flight lines of F11 - F14 in a figure. Otherwise, by words alone, readers feel hard to imagine.

L6L16-17
Does it mean something for readers to know gravity in this paper here?

Section 4.1
I feel there are too much technical terms such as "Waypoint Inertial Explorer", "Precise Point Positioning (PPP)" or "the SPAN IMU". It seems too much for readers who are just interested in candidate locating of ice coring. Please provide explanations more here or in the appendix.

P7L4-5
*Internal estimates of uncertainty for these data have 2 cm height standard deviation.*
The authors seem to tell that 2 cm is for height. How about uncertainty for horizontal positioning?

L7L10-13
*The data was then processed using the "1-D" focused SAR approach of Peters et al.(2007), where focusing of the along track Doppler phase variations within each range resolution cell was employed to improve* *the along track resolution to approximately 10-20 meters for scattering targets.*

Meaning is unclear to me. Do you mean that the processed data contain information over 10-20

meters along the flight line?

*The data was resampled to 4 Hz along track sampling (~22 m) for manual interpretation.*

Here again, meaning is unclear to me. What is your original sampling frequency. In think you did not tell it to readers. Do you mean that the aircraft fly ~88 m in a second along the flight line? It is hard to understand.

P7L18
The authors did not apply firn correction. Why? The authors gave systematic error of ~15 m to the ice thickness data by ignoring the correction. How did the authors consider it when compiling with the other data? How can it be compatible with your effort to use the SAR processing?

P7L21
Please explain more about the "first return policy" for readers of non radar expert. As I wrote at Figure 3, this policy will give a bias of underestimation for ice thickness. The policy means that when faint echo appear from the mountains far from the nadir, the faint mountain is considered as thickness from the nadir. A policy of the strongest echo seems better to me. The strongest echoes are most probably return from the nadir.

Figure 4
Apologizing to authors, I strongly feel that this figure 4 is not very important. Just 4 points show large differences mostly because of combined effects of the first return policy (causing a bias to the ice thickness) and the along track SAR processing effect. If the roughness is evaluated like I commented at Figure 3, it seems much more meaningful.
When there are steep slopes, ice thickness data are disturbed because of the footprint. In case of this paper, shape of the footprint is just asymmetric along the track (short) and across the track (long). It does not seem good indicator of bed roughness.

P8L6
"the critical angle of refraction"
Do you mean Brewster's angle? If so, please add words. Then, more readers will understand. Do the antennas have beams wider than Brewster's angle of 34º? Please clarify this point, too.

Figure 5
To see this figure, I am afraid the authors' wording "Northings" and "Eastings" in many figures are wrong, confusing us.

P9L1
The underestimate is because the authors chose the first echo for determination of the ice thickness.

P9L3
The author' claim here is not clear to me, to see Figure 5, there is no clear tendency.

P10L15-16
I do not find any convincing tendency that authors are claiming here for Figure 6. Just four points like we saw in Figure 4 show deviation due to the effective asymmetric shape of the radar footprint.

Section 6
P11L7-9

If the authors compile the data with old unpublished data, please provide at least a list of comparison for data processing and radar specifications. In addition, a map showing the locations of measurements should be given. The authors' data probably have some bias. How are various sets of data compatible with each other, to be ready to compile together?

What about crossover errors between sets of data?

Section 6 in general
It is really hard for readers to understand what the authors are discussing in the figure. Please provide links between description and indications in figures. Otherwise, descriptions do not mean much. The authors sometimes mention subglacial lakes. However, without demonstration of lakes in figures, readers feel really hard. Please provide a figure showing distribution of lakes and subglacial water nicely. In addition, I felt confused by description of directions in this paper.

Conclusion
2. Why is the candidate A promising? Did it pass all the conditions given in the introduction? "A large number of subglacial lakes" are not informed in this manuscript. What is "distinct basal ice"?

Publication of the data
I suggest all the ice thickness data used in this paper should be published as supplementary data of this manuscript.

Publication of the DEM
I suggest all the ice thickness map given in this paper such as Figure 7 should be published as supplementary data of this manuscript.

The paper will be much better if distribution of subglacial lakes and dry/wet distinction is mapped

Figure 8
Again, background MODIS data means nothing for readers because it is just grey.

---

## Referee Comment (RC2) · Anonymous Referee #2 · 7 Oct 2016

Review of: High resolution boundary conditions of an old ice target near Dome C, Antarctica, Young and others, TCD doi:10.5194/tc-2016-169, 2016

This paper largely concerns the presentation of new aerogeophysical data acquired in the vicinity of Dome C, Antarctica, as part of a concerted international effort to locate a site for coring ice older than 1 Ma. It begins by giving some background context on the "search for the oldest ice," and justifies well the acquisition of aerogeophysical survey data across the Dome C region to this purpose. The paper then describes the survey that was conducted across the region in 2015/16 in response to this need, and presents an impressive new map of the bed, as well as incorporating a valuable discussion of the uncertainties of the data.

The paper certainly presents data that are worth publishing and that will be of high interest to a wide range of readers, most obviously in the ice coring community but also more generally – for example radioglaciologists and geomorphologists will find the analysis of radar uncertainties valuable. Not for the first time, the ICECAP team have acquired a hugely impressive product in a highly remote part of Antarctica; a highly commendable undertaking. The figures are produced to a high standard, albeit I suggest some amendments in my specific comments.

However, there are two aspects to the paper which I recommend need major attention:

Most importantly, there is too little analysis/interpretation/discussion of the data. In its current form, and not including the material on uncertainty analysis (which I think is misplaced in the structure; see next comment), the absence of a discussion of the data renders the paper little more than a dataset presentation paper, possibly more suitable for the journal Earth System Science Data which publishes datasets without the requirement for detailed analyses. Given the paper's set-up that the survey was undertaken to help site the possible new ice core, the fact that it doesn't offer a detailed reflection on the new insights it has contributed towards honing in on the oldest-ice coring site seems a big miss. Fundamentally the paper just closes down too rapidly once it reaches the results section (just as I was getting most interested….). I also think the bullet point conclusions section needs consideration: I make some suggestions about this in my Major Comments below.

Secondly, and this is a lesser, but related, comment to the above, the paper has some structural problems. The clearest example of this is Section 5, the uncertainty/crossover discussion: this is actually a very valuable aspect to the paper but comes across as rather ad hoc because it is written entirely within an isolated section and not treated as part of the overall methods/results/[discussion]. I also suspect its importance will be overlooked by any but the most thorough readers because the abstract/conclusions only say the issue is investigated in the paper, rather than summarising the findings. There are also a few places where a reordering of figures might help.

Thus – for the importance of the topic and the clear worth of the data, I would really like this paper to progress, but I feel it needs some reworking as regards the presentation, especially with regards to closing the loop in terms of what the data now offer in terms of identifying an "oldest-ice" site. In its current form, the paper significantly undersells the importance of the data collected and, indeed, the efforts that have gone into collecting it.

**Major comments**

My main comment is that I think the paper needs some restructuring and expansion from Section 5 onwards (and then reflecting this in a revised and more detailed abstract).

Most importantly, I think that the paper requires a Discussion section between the sections currently entitled "6. Results" and "7. Conclusions." There is clearly lots of potential to do all sorts of exciting further analyses of the new aerogeophysical data collected for this study, and I do not advocate the authors do more analysis per se (presumably they have plans for follow up papers on the data) – BUT I do think they can make some more of the dataset they present here already. Can they present an assessment as to whether the new data have helped determine whether or not Candidate Site A is a better/worse candidate site than was thought before the 2016 survey? I think there is also some potential to explore a little further how the issues surrounding different uncertainties/errors for datasets of different provenance are dealt with in producing the combined product Figure 7. It would be valuable to see just the data acquired from 2016 presented as a bedmap in isolation of the other datasets, as well as the bedmap produced with the combined datasets.

I also recommend that the authors consider how they can reframe all the material in Section 5 into relevant methods/results/discussion sections (the fringe benefit is that some of this material can essentially form one section of the Discussion the paper so desperately needs).

I recommend the authors then rewrite/revisit their overall conclusions and abstract so that they state in both of those sections the findings that result *directly* from the new dataset and analysis. The current conclusion 3 especially only states a fact that the authors tell us they already knew before they collected the new data presented here.

**Minor comments**

The above major comments now aside, I would like to emphasise that I found most of the paper well written and the figures of good quality. I don't think the authors need to do much to the opening of the paper, as they may infer from the selection of comments below.

Minor suggested global edits:

There should be no apostrophe in 1970s, 1990s etc.

Is there any need for the acronym CST for Concordia Subglacial Trench? You don't use CST that often (unless an expanded Discussion will use it much more) so it just seems an unnecessary acronym.

Introduce hyphens into "along-track, "across-track," "off-track," "range-compressed," "pulse-limited"…

Ensure "crossover" consistently expressed as one word.

Sections 1 and 2:

P2, L1: Change "criteria" (plural) to "criterion" (singular).

P2, L4: I suggest you don't need the aside about these features being called "blobs." You don't refer to blobs elsewhere in the paper.

P2, Section 1, final paragraph: The current wording is vague about whom the "European-led group" are, and includes some extraneous detail about the logistical delay to the survey - you have my sympathy on the latter, but it doesn't affect the findings of the paper. This paragraph just has to focus on the purpose of this paper, which I suggest is along the lines of: "In this paper we present the results of an aerogeophysical survey specifically targeting the candidate old-ice access sites that was conducted in January 2016. We show … [now state what the paper fundamentally shows and adds to existing knowledge, e.g, a new map of the basal topography, and preferably one or two ways in which you use the data more than just

presenting a map, i.e. what do the new data add to identifying an old-ice access site? As a further example of the new data's uses you could also state that the new data offer insight into the meaning of uncertainties in RES data analysis.]

Figures 1 and 2; wherein currently Figure 1 is first referenced within the text in the opening line to Section 2, essentially just to locate Dome C:

To aid overall readability of Section 2 and the inevitable flicking between text and figures, I think you could combine Figures 1 and 2, and add a new panel, so that the figures are more readily intercomparable by the reader and introduced more logically as the material is discussed in the text. Essentially, following the text, the first result I want to see is just the surface ice topography (as per Section 2.1, paragraph 1), then I want to see the pre-ICECAP-surveyed subglacial topography (as per Section 2.1, paragraph 2), then I want to see the Van Liefferinge model results (as per Section 2.2). So I'd suggest a new three-panel Figure 1 covers the bases in that order, i.e. panel (a) shows pre-ICECAP surface topo; (b) shows pre-ICECAP subglacial topo, and panel (c) shows Van Liefferinge results. It would be really useful if every panel had superimposed the 5x candidate ice-core sites from Van Liefferinge as well as Concordia. The figure would also need still to include an inset showing the general location of Dome C. It would also be valuable on at least one panel (subglacial topo perhaps) the locations of subglacial lakes from the latest inventory marked.

W.r.t. the existing caption for Fig. 1, the acronym OIA is unexplained in the main text at the stage I first read the caption, and I suggest the sentence referring to the red line is reordered as: Red line shows radar profile acquired in 2011 and shown in Figure [would now be 2, if you follow my suggestion to combine Figures 1 and 2].

A minor point on the existing Fig. 2 – the blue contour line to the left, presumably denoting surface elevation 3200, could do with labelling within the figure.

P2, L20: At end 1st sentence of this section, just point reader to relevant figure showing surface topo.

P2, L20: Change "was" to "were" and explain acronym "INS."

P2, L22, 31 & 32: In these contexts, no need for "dome" to have capital "D."

P2, L25: No need for phrase about Dome C Lake District.

P2, L32: At end sentence "…northward flow" point reader towards relevant figure (currently Fig 2, but as per above comment suggest this becomes Fig 1b).

P3, L1-2: The sentence introducing the "broad channels" doesn't make clear whether the broad channels are surface features, bed features, or possibly both, and it would be improved if some idea of the dimensions of the relevant features were included. Could the authors consider showing these features explicitly in my suggested new panel Fig. 1a?

P3, L9-14: There's some unnecessary information here about surveys which aren't used for this paper. I also suggest, for structural reasons, that you introduce the 2011 data in Section 2.2 (see comment below). Thus I think you could just excise these lines.

P3, L16: Remove "have".

P3, L17: sp. teleseismic

P4, L3: change to "[none]….sites *overlaps* with *the*…"

P4, L13: Here is where I think you could say, for the first time, that ICECAP/HiCARS2 profiled across the Candidate A site in 2011. I suggest you also find alternative wording to your use of "core" in the current sentence.

Figure 3: Just a minor point – why have distance going from right to left? Intuitively it would just seem preferable to have this axis reversed, if only to adjoin the description of the englacial layers diving off a cliff as mentioned in the main text. Admittedly this is not a major issue.

In the caption to Figure 3: typo: "along" rather than "alone". The caption should also mention that the profile location is shown on Figure [1c…?].

Sections 3 and 4

I suggest that both of these sections essentially outline the "Methods" or "Methodology" and could be titled as such in a single section.

I think Section 3 misses an opening sentence or two to remind and re-orient the reader that you are now going to focus on data collected in 2016.

Table 1, Row 2: sp. Scalar

You introduce/describe in this section and list in the table some instruments whose data are not apparently used at all within the paper.

P6, L4: reverse order: "constrain better"

P6, L8-9: Unnecessary and could just be cut.

P6, L12: "…helped *to* refine…"

P6, L13 and throughout Section 4.2: "…data *were*…"

P7, L9: "…[hyperbolae]….characterize…" i.e. not characterizes with an "s"

P7, L17: Move comma: "…column and, using…"

P7, L18: Reverse "not" and "to"…. We choose not to apply…"

I'm surprised to see no mention of the assumed radiowave travel speed in Section 4.3. In general, though, I completely approve of the no frills approach to the data processing in the paper.

I do not provide minor comments from Section 5 onwards following my view that they require major revision.

---

## Referee Comment (RC3) · Anonymous Referee #3 · 11 Oct 2016

**Review:** "High resolution boundary conditions of an old ice target near Dome C, Antarctica"
Duncan A. Young, et al., *The Cryosphere*

**Summary:**

This paper presents the results of a densely spaced ice-penetrating radar survey near Dome C in East Antarctica. The survey connects to the location of the oldest ice core record available at the present day, EPICA Dome C, although the main body of the survey is displaced somewhat relative to that core. The survey was conducted for the purpose of evaluating potential drill sites for a new oldest ice core. The hope is that the new ice core will recover stratigraphically intact ice dating to 1.5 Ma, before the mid-pliestocene transition when the Earth's climate system switched from 40 ka ice age cycles to 100 ka cycles. Such a core would represent a substantial improvement over the 800 ka record recovered from EPICA Dome C.

The authors identify five candidate locations (A-E) based on regions identified as cold-based in previous modeling work. The authors find that no drill location is perfect and make no explicit endorsement of any individual drill location. However, Candidate A receives a much higher proportion of their attention and analysis, and the paper therefore reads as implicitly endorsing Candidate A. Candidate A is by far the largest cold-based region in the model, and the survey grid is designed to be densest over Candidate A. The authors find that Candidate A has rugged basal topography and isolated subglacial lakes likely to cause stratigraphic disturbances in the basal ice, and the authors also observe some evidence of this disturbance in the form of a basal ice unit. Nonetheless, the authors find that Candidate A has "some promising sites" where conditions are favorable to recovering an old ice core.

**Major Comments:**

My biggest criticism of this paper is that it lacks interpretation and discussion of the geomorphology or glaciology of the survey area. The paper reads as a limited technical report of a site survey without much scientific interpretation. Large portions of the results are presented in what is essentially an extended methods section, while the actual results section is barely half a page long and a discussion section is completely absent. *This paper needs a discussion section.* Section 5 briefly alludes to fractal measurements of landscapes, but this analysis is not expanded on in any way. The readers are left wondering, "so what?" The scientific quality of this paper can be greatly improved by including analysis of the subglacial landscape and overriding ice sheet. For example, the authors could interpret the landscape in terms of physical processes, such as subglacial erosion versus preglacial erosion, or tectonic deformation. The authors could present the locations of the subglacial lakes that they allude to, and discuss the relationship between their findings and the extensive literature on subglacial lakes that already exists. Either of those possibilities, or other possibilities that the authors feel would be more appropriate, would help to place the paper within a broader scientific context and expand it's appeal beyond the ice core community.

In addition, the authors should do a better job of owning their preference for Candidate A. Candidate A was the primary target of the survey and received by far the most attention in the paper. I think that this preference is justified- as I mention below, the other candidates are only a few grid cells wide and therefore cannot be considered reliable predictions of the numerical model- but the authors need to explicitly own this preference. The conclusions mention "some promising sites" within Candidate A, but the paper spends little if any time exploring these sites. The results section could

easily be expanded to investigate several sites within Candidate A; call them Candidate A1, Candidate A2, etc. This would move the ice core conversation forward, from "which candidate?" to, "where specifically within Candidate A are we going to drill?"

**Minor Comments:**

Note on line numbering: The line numbers restart at the beginning of each page, rather than counting from the start of the document. I therefore include both a page number and a line number in my comments.

P 1, L 3-4: "We find under the primary candidate region elevated rough topography, near a number of subglacial lakes, but also regions of smoother bed."
This wording is awkward and requires several readings to understand. One possible rephrasing is, "We find that the primary candidate region contains elevated rough topography interspersed with scattered subglacial lakes and some regions of smoother bed."

P 1 L 15-18: requirements for an intact ice column
Requirements 1 and 5 (low geothermal flux and low ice thickness) are really part of the same requirement: that the ice must be cold-based. Cold-based conditions require that the geothermal (and frictional) heat flux be low relative to the conductive heat flux, which is inversely proportional to ice thickness. The stated threshold of 2500 m is really a function of the geothermal flux. In addition, the cold-based requirement is in conflict with the low accumulation requirement, as lower accumulation rates tend to produce a warmer ice column and higher accumulation rates produce a colder ice column. A sentence or two outlining the physics behind these requirements would be useful here.

Section 2.2:
Mention that Candidate A is favorable because it is the largest candidate. The other candidates are only a few grid cells large, and are therefore unreliable. The thermomechanical model used to define the candidates is a continuum model, and therefore cannot be expected to accurately describe features on the grid cell scale. Candidate A is the only candidate that is much larger than the grid size, and is therefore the only candidate that can be considered a robust prediction of the model. This is actually the most powerful argument in favor of Candidate A.

P 4 L 6: "...basal ice likely traverses the..."
The basal ice traversed the trough in the past, replace with "has likely traversed".

P 4 L15 – P 5 L2: "...while in the bottom 500 m, a region of more diffuse englacial scattering is present. This distinct zone of basal ice is also apparent in McCoRDS radar data that operates at a higher frequency."
This is a good place to reference Bell et al., 2011. The diffuse englacial scattering is similar to what they described as "valley wall" accretion ice near Dome A.

P 7 L21: "We...maintain a strict first return policy."
In an area of rough basal topography, there is a good chance that the first return may come from off-nadir bed returns. In fact, this is almost certainly what happened, given the results of Section 5. It might be good to include a sentence here mentioning that this first return policy likely resulted in picking off-nadir returns as the bed, and that you explore this in greater detail in the next section.

Section 5:
A large amount of the material in this section would be more suitable for the results section.

P 10, L5-12:
Why not compute H for this dataset (or for the subset of this dataset within Candidate A)?  You could determine how RMS roughness varies as a function of window size, and perhaps use this information to say something about the processes responsible for shaping the landscape.  This goes to my major comment above.

P 10, L13:  "Figure 6 shows the relationship between RMS deviation at 1600 m length scale..."
The axes label of Figure 6 says 800 m length scale.

P 10, L15-16:  "A stronger relationship is seen for the focused data than for the pik1 data, primarily due to the larger crossover differences seen in the focused data."
First comment:  the sentence would be clearer if you said "...seen for the focused data than for the unfocused data..." rather than using code ("pik1").
Second comment:  The second half of this sentence would be more compelling if you said "primarily due to the geometric arguments given earlier".  The crossover differences are larger for the focused data than for the unfocused data because the unfocused data includes off-nadir returns in both the along-track and across-track directions, but the focused data only has off-nadir returns in the across-track direction.

**Figures**

The map figures need to have some indication of latitude and longitude.

Figure 1:
Specify in the caption that the "candidates" refer to the cold-based regions.  I was looking for specific dots on the map.

Figure 2:
The 10 m contours are extremely difficult to see in a printout.  The inset map of Antarctica would be better suited for figure 1.  Overlay the boundary of Candidate A.

Figure 3:
Add more x-axis labels (say, every 10 km).  Put the units (dB) on the colorscale.  Indicate the boundaries of Candidate A.  Add a note to the top left or bottom left corner of the image indicating the direction to Dome C.
It might also be helpful to show the echogram going all the way to Dome C.  This will allow the x-axis scale to begin at zero, and (more importantly) it will allow the reader to assess how the continuity of the internal layers in Candidate A compares with the continuity of the internal layers at Dome C.  If the echogram is expanded this way, you should also add a vertical line indicating the location of the Dome C ice core (or the closest approach to the core), with a tick indicating the lowest depth from which stratigraphically intact ice was recovered.

Figure 4:
It might be helpful to show another set of histograms where the range has been truncated at ±100 m, so that the scale is not distorted by a few large outliers.

Figure 5:
It is hard to see both the crossovers and the bed elevation, as both have similar color scales. It might be better to have the bed elevation in black and white. Alternately, it might be good to have two separate panels, one showing the bed elevation and one showing the crossovers. Consider merging figure 7 with this figure in that case.

Figure 6:
It is hard to tell the two colors apart, and most of the figure space is blank white space. Consider splitting foc1 and pik1 into separate subplots. Also, consider using log-log axes to more efficiently use all of the space. In addition, the caption says that the RMS window was 1600 m, while the axis label says it was 800 m.

Figure 7:
See my note above about potentially merging this figure with figure 5.

Figure 8:
Again, was the lengthscale 800 m or 1600 m?
The MOA background adds nothing to the figure, as it is a uniform gray. Consider using Bedmap2 as the background, with the same grayscale as the new bed elevations.

---

## Author Comment (AC1) · 17 Jan 2017

Please see the attached response zip file, which contains supplementary data, the revised manuscript (PDF) and the response to the reviewers (PDF)

Please also note the supplement to this comment:
http://www.the-cryosphere-discuss.net/tc-2016-169/tc-2016-169-AC1-supplement.zip

---

## Author Response (AR1)

**Response to reviewers on "High resolution boundary conditions of an old ice target near Dome C, Antarctica" by Duncan Young et al.**

We thank the reviewers for their advice; the paper is now greatly improved, with a much greater focusing on the motivating old ice search. The discussion section has been added, and the results section expanded. Subglacial lakes are included in detail, as requested by a couple of the reviewers. However, englacial reflector interpretation is outside the scope of this paper. A large section in the original paper focusing on cross over analysis has been migrated to an appendix.

**Reviewer 1**

Reviewer 1 wished this paper to be more focused on ice core scientists, which is appropriate give the target volume. We expanded the suggestion that we focus more on subglacial water; however, englacial reflector mapping and modeling is outside the scope of this paper, with followup papers dealing with those subjects in progress. We also at the reviewer's recommendation expanded our discussion of the compilation of the datasets.

**Specific comments:**

**Abstract: It is too concise with about 100 words. Please use space of the abstract efficiently. Please use much more words and space (2~3 times of present length) to explain what are really new in the manuscript in terms of science…**
This has been expanded, and subglacial water has been added to the manuscript

**P2L6: I did not find candidates A - E in cited papers. Please explain.**
This paper now claims the A-E nomenclature on page 5, line 6. "*In the Dome C region, five candidate sites exist, which we term A, B, C, D, and E.*"

**P2L9-10: Please provide citations for ICECAP and ICECAP2.**
 A citation is now provided for ICECAP. This paper is the first publication from ICECAP II

**Section 2: Please provide a map showing entire Antarctica and Dome C region in it. In addition, please provide a map with site locations such as Totten Glacier, Byrd Glacier or George V Coast, VCD corridor…**
A new Figure 1, showing East Antarctica is provided with the key sites referenced in the text

**P2L20: "crude" -> "pioneering"? …**
Crude is changed to pioneering

**P2L21: Why "however"? Did somebody question presence of dome?**
In this expanded version of the paper, we make a stronger point on the uncertainties in the older data.

**Figure 1 Caption "OIA" appeared here without any explanation before. Background MODIS data has no contrast within it. Thus there is no meaning to show it here. "Antarctic Polar Stereographic" Please provide standard latitude.**
We move up the introduction to the OIA survey, we replace MODIS, and we describe the standard latitude.

**Figure 1 Figure Please indicate X and Y directions because they are often mentioned in the text. In addition, horizontal axis is used as Y by the authors.**
This figure (now merged with figure 2) is greatly expanded. We indicate the directions in supplementary material.

**Section 2.1: Please provide much more information of the radar system used for this work, such as peak power, pulse compression rate, antenna gain, beam (half power) width in E direction and H direction, effective thickness resolution in ice. Perhaps it may be found in the Peters paper or the other papers. However, it is not kind at all for the authors not to show concrete information to readers in this paper. What is "focusable data"? Please explain to readers kindly and properly.**

We now describe relevant parameters in more detail in section 3.4. Peak power and the details of the beam pattern do not affect the conclusions of this paper.

**P3L14: What is Vostok/Concordia/DDU corridor (VCD)? Please explain using a figure. What is DDU? Please think about readers who are not familiar to this region.**
This is expanded in Figure 1

**Figure 2: Please indicate area A - E once again in this figure. Otherwise, readability is not good. Please indicate flight lines of Figure 1 once again on this map to make better readability.**
Candidates A-E are now indicated on all maps of the region of interest.

**P4L4-5: "Sites B, C, and D are located on the steep and poorly sampled peaks on the northern side of the Concordia Subglacial Trench (CST)"; Does it mean bed or surface? B, C and D seems on the southern side of the CST in Figure 2. Am I wrong? Then why?**
Rewritten for clarity: "*Sites B, C, and D are located on the steep and poorly sampled subglacial peaks on the northeastern side of the Concordia Subglacial Trench*"

**The authors seem to use too much symbols such as DDU, CST, VCD and so on. It seems too much for readers. Later in the manuscript, I felt hard to remember their meanings. Where is CST on the map? It is hard for readers who are not specialist for this region.**
Additional text (P3L4) now indicates the location on the Concordia Subglacial Trench "The coarse subglacial geography revealed by the Italian survey comprises of a deep subglacial trough (the Concordia Subglacial Trench) to the northeast of Dome C (see lower left of Fig. 2 a)"

**P4L6: "basal ice in this region likely traverses the deep, wet CST and is unlikely to be stratigraphically intact." t seems still a vague guess. Mountainous area has at least width of ~10 km. Can you suggest some direct/ indirect evidence, for example, internal layers?**
The language was softened by eliminating the last clause. Analysis of the internal reflectors is the subject of a followup paper.

**P4L11-12: "The ice surface above Candidate A forms a topographic extension to the south of Dome C informally termed 'Little Dome C'. The central part of Candidate A lies 40 km south from Concordia Station." It seems south and north are very confusing in this manuscript and maps, like X and Y. Please make them very clear to readers.**
We have added north arrows to the figures, and Figure 1 should help orient the reader.

**P4L13: VCD/JKB2g/DVD01a is a kind of jargon for readers.**
The line name is a necessary index into the dataset, which is in the process of being released at NSIDC.

**P4L13-14 "Focusing of the radar data showed that the southern flank of the Candidate A massif ended in a steep cliff over which englacial layers dive." First, I was confused in terms of directions. Second, "dive" seems inadequate because the authors used very high vertical exaggeration of ~20 times in Figure 3. If the authors use real scale, it should be very smooth, flat and continuous layers. "dive" is just an artifact effect that the authors produced by exaggeration.**
Figure 3 a is now redone at true scale, and the englacial reflectors still noticeably dip over this scarp. Directions are clarified.

**Figure 3: Did the authors apply the geometrical spreading effect in dB? Or, are these data just return power from targets? Please make this point clear for readers. Please indicate south and north.**
*The geometric loss correction is clarified, and distance from Concordia is explicitly shown.*

**Please indicate this segment of the flight line on the maps (Figures 1 and 2).**
*This line is now shown on the combined Fig 2.*

**This figure is given but not discussed in the manuscript.**
This figure is discussed in two locations in the text: Section 2.2 and Section 2.3.2

**Figure 3 I suggest that roughness and amplitude/frequency of it should be analyzed using such data here. It is much better data source compared to the crossover differences or RMS that the authors are discussing in this manuscript. For example, at a site of X=50 - 55 (km), difference between the first echo and the strongest echo is as large as 200 m or more. By choosing only the first echo for ice thickness analysis, the authors ignore the strongest echo which is most probable echo from the nadir. With this reason, the authors analytical produce is causing a bias of underestimation of ice thickness from the beginning. In addition, analysis of the both leading and trailing edges will give very good measures of the bed roughness.**
We have to be careful here, for X=50-55 the difference is not really a vertical distance 200 m, it is a delay ~2.5 μsec, with a large cross track and smaller vertical component.   The methodology suggested by this comment is not stable in the presence of complex geometry, with apparent ice thickness jumping hundreds of meters between traces as first the lower then the higher echo trade being the brighter echo, and would serve to exaggerate the along track roughness. Exactly this effect is seen in the older Italian data, which we comment on in this paper.
Analysis of the leading and trailing edge does not quantify roughness without a complex scattering model for the bed which is outside of the scope of this paper.  It is also sensitive to SNR considerations (ie, if a bed echo is close to the noise floor, the trailing edge will be truncated, and thus appear "smoother").  For a range compressed system like HiCARS, the leading edge has no physical meaning.  The lead author has employed trailing edge analysis (Young et al., 2016, PTRS), however in very specific circumstances to evaluate specific hypothesis.

**P5L7 What is Internal Measurement Unit? Please explain to readers.**
This is now explained in Table 1.

**Title of the section 3 Explanation of the OIA is given only later. Please explain to readers.**
OIA is now explained in Section 2.4.

**Table 3 This seems a strange table to me with several reasons. (i) Why commercial names of the instruments appear in the first column? Item of measurements should be given first such as ice thickness, distance between aircraft and the ice sheet, position, etc. (ii) F11 - F14 are not given well in this manuscript. It is hard to understand. Only the authors know well.**
As suggested, we added the measured parameter to the table.  We change the flights to dates; however, we feel some metadata must be provided to give readers informed access to the data.

**(iii) Please give priority to instruments that you really used for discussions of this manuscript. Gravity and Geomagnetic are not discussed in this paper. Radar sounder and positioning should be shown with higher priority here.**
MARFA was already at the top of the list; we move the laser up as well.

**(iv) What is ICECAP2? What is different from ICECAP? Few readers know them.**
We change this to the OIA instrument suite.

**P6L4-5 "Flight lines were designed to avoid Concordia's clean air sector to the south of the station, as well as to allow the aircraft to make VHF communications with the station before landing." I suggest this should be removed because it is something that only very limited people should know.**
This language was added to inform anyone who wishes to perform a follow up investigation.

**P6L14 "Elevation difference" Does it mean elevation of the ice sheet surface?**
We clarify to "Apparent surface elevation differences between survey lines"

**Section 3.2 Please prepare a figure showing flight lines of F11 - F14 in a figure. Otherwise, by words alone, readers feel hard to imagine.**
This is added in supplementary data

**L6L16-17 Does it mean something for readers to know gravity in this paper here?**
This line was deleted.

**Section 4.1: I feel there are too much technical terms such as "Waypoint Inertial Explorer", "Precise Point Positioning (PPP)" or "the SPAN IMU". It seems too much for readers who are just interested in candidate locating of ice coring. Please provide explanations more here or in the appendix.**
This section is consolidated within the Methods section, and was designed to appeal to those interested in high resolution aerogeophysical surveys.

**P7L4-5: Internal estimates of uncertainty for these data have 2 cm height standard deviation. The authors seem to tell that 2 cm is for height. How about uncertainty for horizontal positioning?**
An estimate for horizontal uncertainly is added.

**L7L10-13 The data was then processed using the "1-D" focused SAR approach of Peters et al.(2007), where focusing of the along track Doppler phase variations within each range resolution cell was employed to improve the along track resolution to approximately 10-20 meters for scattering targets. Meaning is unclear to me. Do you mean that the processed data contain information over 10-20 meters along the flight line? The data was resampled to 4 Hz along track sampling (~22 m) for manual interpretation.**
This section has been expanded and clarified.

**P7L18 The authors did not apply firn correction. Why? The authors gave systematic error of ~15 m to the ice thickness data by ignoring the correction. How did the authors consider it when compiling with the other data? How can it be compatible with your effort to use the SAR processing?**
We add the following: "We choose to not apply a firn correction to ice thicknesses; as shown in Peters et al., 2007, a firn correction is not required for our focusing, and and will not affect the conclusions in this paper (firn correction is however critical for isochron interpretation)"

**P7L21 Please explain more about the "first return policy" for readers of non radar expert. As I wrote at Figure 3, this policy will give a bias of underestimation for ice thickness. The policy means that when faint echo appear from the mountains far from the nadir, the faint mountain is considered as thickness from the nadir. A policy of the strongest echo seems better to me. The strongest echoes are most probably return from the nadir. Figure 4 Apologizing to authors, I strongly feel that this figure 4 is not very important. Just 4 points show large differences mostly because of combined effects of the first return policy (causing a bias to the ice thickness) and the along track SAR processing effect. If the roughness is evaluated like I commented at Figure 3, it seems much more meaningful.**
**When there are steep slopes, ice thickness data are disturbed because of the footprint. In case of this paper, shape of the footprint is just asymmetric along the track (short) and across the track (long). It does not seem good indicator of bed roughness.**
**P8L6 "the critical angle of refraction" Do you mean Brewster's angle? If so, please add words. Then, more readers will understand. Do the antennas have beams wider than Brewster's angle of 34o? Please clarify this point, too.**
This whole section has been condensed and moved to an appendix. Figure 4 has been removed.

**Figure 5 To see this figure, I am afraid the authors' wording "Northings" and "Eastings" in many figures are wrong, confusing us.**
Figure 5 is deleted. Northings and Eastings have been used to describe the x and y axis of the Antarctica Polar Stereographic projection, including by NSIDC, the Polar Geospatial Center, the Australian Antarctic Data Center, and SCAR in its formal definition of the Antarctica Polar Stereographic projection. We change this to Projected Northings and Projected Eastings, and add a north arrow to all plots.

**P9L1 The underestimate is because the authors chose the first echo for determination of the ice thickness.**
**P9L3 The author' claim here is not clear to me, to see Figure 5, there is no clear tendency.**
**P10L15-1 I do not find any convincing tendency that authors are claiming here for Figure 6. Just four points like we saw in Figure 4 show deviation due to the effective asymmetric shape of the radar footprint.**

This section has been edited back and put in the appendix.

**Section 6 P11L7-9 If the authors compile the data with old unpublished data, please provide at least a list of comparison for data processing and radar specifications. In addition, a map showing the locations of measurements should be given. The authors' data probably have some bias. How are various sets of data compatible with each other, to be ready to compile together? What about crossover errors between sets of data?**
Following on this suggestion, this section has been greatly expanded in Section 3.7.

**Section 6 in general It is really hard for readers to understand what the authors are discussing in the figure. Please provide links between description and indications in figures. Otherwise, descriptions do not mean much. The authors sometimes mention subglacial lakes. However, without demonstration of lakes in figures, readers feel really hard. Please provide a figure showing distribution of lakes and subglacial water nicely. In addition, I felt confused by description of directions in this paper.**
This section has been greatly expanded

**Conclusion**
**2. Why is the candidate A promising? Did it pass all the conditions given in the introduction? "A large number of subglacial lakes" are not informed in this manuscript. What is "distinct basal ice"?**
This section has been greatly expanded.  The reference to basal ice has been deleted.

**Publication of the data I suggest all the ice thickness data used in this paper should be published as supplementary data of this manuscript.**
**Publication of the DEM I suggest all the ice thickness map given in this paper such as Figure 7 should be published as supplementary data of this manuscript.**
These data will be provided as supplementary data.

**The paper will be much better if distribution of subglacial lakes and dry/wet distinction is mapped**
We have greatly expanded our discussion of subglacial water

**Figure 8 Again, background MODIS data means nothing for readers because it is just grey.**
Figures 7 and 8 have been folded into a new Figure 5.  MODIS has been removed from all figures.

**Reviewer 2**

Reviewer 2 addressed the lack of analysis/interpretation/and discussion in this paper, and the ad hoc structure of the paper. We addressed this with a large scale reorganization and expansion of discussion of the relevance for old ice.

**Minor suggested global edits:**
**There should be no apostrophe in 1970s, 1990s etc.**
Done

**Is there any need for the acronym CST for Concordia Subglacial Trench? You don't use CST that often (unless an expanded Discussion will use it much more) so it just seems an unnecessary acronym.**
Done

**Introduce hyphens into "along−track, "across−track," "off−track," "range−compressed," "pulse−limited"... Ensure "crossover" consistently expressed as one word.**
Done

**Sections 1 and 2: P2, L1: Change "criteria" (plural) to "criterion" (singular).**
Done

**P2, L4: I suggest you don't need the aside about these features being called "blobs." You don't refer to blobs elsewhere in the paper.**
Done

**P2, Section 1, final paragraph: The current wording is vague about whom the "European−led group" are, and includes some extraneous detail about the logistical delay to the survey − you have my sympathy on the latter, but it doesn't affect the findings of the paper. This paragraph just has to focus on the purpose of this paper, which I suggest is along the lines of: "In this paper we present the results of an aerogeophysical survey specifically targeting the candidate old−ice access sites that was conducted in January 2016. We show ... [now state what the paper fundamentally shows and adds to existing knowledge, e.g, a new map of the basal topography, and preferably one or two ways in which you use the data more than just presenting a map, i.e. what do the new data add to identifying an old−ice access site? As a further example of the new data's uses you could also state that the new data offer insight into the meaning of uncertainties in RES data analysis.]**
We focus this section, and follow up on the recommendation to reorder and expand the discussion and conclusions

**Figures 1 and 2; wherein currently Figure 1 is first referenced within the text in the opening line to Section 2, essentially just to locate Dome C:**
**To aid overall readability of Section 2 and the inevitable flicking between text and figures, I think you could combine Figures 1 and 2, and add a new panel, so that the figures are more readily intercomparable by the reader and introduced more logically as the material is discussed in the text. Essentially, following the text, the first result I want to see is just the surface ice topography (as per Section 2.1, paragraph 1), then I want to see the pre−ICECAP−surveyed subglacial topography (as per Section 2.1, paragraph 2), then I want to see the Van Liefferinge model results (as per Section 2.2). So I'd suggest a new three−panel Figure 1 covers the bases in that order, i.e. panel (a) shows pre−ICECAP surface topo; (b) shows pre−ICECAP subglacial topo, and panel (c) shows Van Liefferinge results. It would be really useful if every panel had superimposed the 5x candidate ice−core sites from Van Liefferinge as well as Concordia. The figure would also need still to include an inset showing the general location of Dome C. It would also be valuable on at least one panel (subglacial topo perhaps) the locations of subglacial lakes from the latest inventory marked.**
We converted Figure 2 into a new composite figure with bedmap, surface slope, old coverage and new coverage, all showing the van Liefferinge (2013) results.

**W.r.t. the existing caption for Fig. 1, the acronym OIA is unexplained in the main text at the stage I first read the caption, and I suggest the sentence referring to the red line is reordered as: Red line shows radar profile**

**acquired in 2011 and shown in Figure [would now be 2, if you follow my suggestion to combine Figures 1 and 2].**
Done

**A minor point on the existing Fig. 2 – the blue contour line to the left, presumably denoting surface elevation 3200, could do with labelling within the figure.**
Done

**P2, L20: At end 1st sentence of this section, just point reader to relevant figure showing surface topo.**
Done

**P2, L20: Change "was" to "were" and explain acronym "INS."**
Done

**P2, L22, 31 & 32: In these contexts, no need for "dome" to have capital "D."**
Done

**P2, L25: No need for phrase about Dome C Lake District.**
Deleted

**P2, L32: At end sentence "...northward flow" point reader towards relevant figure (currently Fig 2, but as per above comment suggest this becomes Fig 1b).**
Done

**P3, L1−2: The sentence introducing the "broad channels" doesn't make clear whether the broad channels are surface features, bed features, or possibly both, and it would be improved if some idea of the dimensions of the relevant features were included. Could the authors consider showing these features explicitly in my suggested new panel Fig. 1a?**
The language has been expanded "however, revealed broad, shallow channels trending north-south within the subglacial plateau region."

**P3, L9−14: There's some unnecessary information here about surveys which aren't used for this paper. I also suggest, for structural reasons, that you introduce the 2011 data in Section 2.2 (see comment below). Thus I think you could just excise these lines.**
These data are used in the final compilation, so we argue for retention of this language.

**P3, L16: Remove "have".**
Done

**P3, L17: sp. teleseismic**
Done

**P4, L3: change to "[none]....sites overlaps with the..."**
Done

**P4, L13: Here is where I think you could say, for the first time, that ICECAP/HiCARS2 profiled across the Candidate A site in 2011. I suggest you also find alternative wording to your use of "core" in the current sentence.**
Done

**Figure 3: Just a minor point – why have distance going from right to left? Intuitively it would just seem preferable to have this axis reversed, if only to adjoin the description of the englacial layers diving off a cliff as mentioned in the main text. Admittedly this is not a major issue.**

Figure 3 is flipped as requested

**In the caption to Figure 3: typo: "along" rather than "alone". The caption should also mention that the profile location is shown on Figure [1c...?].**
Done

**Sections 3 and 4 I suggest that both of these sections essentially outline the "Methods" or "Methodology" and could be titled as such in a single section. I think Section 3 misses an opening sentence or two to remind and re-orient the reader that you are now going to focus on data collected in 2016.**
We expand the end of section 2 (The Dome C region) to provide context, and combine Section 3 and 4 into a Methods section.

**Table 1, Row 2: sp. Scalar**
Done

**You introduce/describe in this section and list in the table some instruments whose data are not apparently used at all within the paper.**
the description of the potential fields instruments is reduced.

**P6, L4: reverse order: "constrain better"**
Done

**P6, L8−9: Unnecessary and could just be cut.**
Removed

**P6, L12: "...helped to refine..."**
Done

**P6, L13 and throughout Section 4.2: "...data were..."**
Done

**P7, L9: "...[hyperbolae]....characterize..." i.e. not characterizes with an "s"**
Done

**P7, L17: Move comma: "...column and, using..."**
Done

**P7, L18: Reverse "not" and "to".... We choose not to apply..."**
Done

**Reviewer 3**
Reviewer 3 felt that this paper needs a discussion section, and that we must own our preference for Candidate A. We have substantially enhanced the discussion in terms of the hydraulic and glaciological context for Candidate A.

**Minor Comments:**
**P 1, L 3-4: "We find under the primary candidate region elevated rough topography, near a number of subglacial lakes, but also regions of smoother bed." This wording is awkward and requires several readings to understand. One possible rephrasing is, "We find that the primary candidate region contains elevated rough topography interspersed with scattered subglacial lakes and some regions of smoother bed."**
Adopted

**P 1 L 15-18: requirements for an intact ice column**
**Requirements 1 and 5 (low geothermal flux and low ice thickness) are really part of the same requirement: that the ice must be cold-based. Cold-based conditions require that the geothermal (and frictional) heat flux be low relative to the conductive heat flux, which is inversely proportional to ice thickness. The stated threshold of 2500 m is really a function of the geothermal flux. In addition, the cold-based requirement is in conflict with the low accumulation requirement, as lower accumulation rates tend to produce a warmer ice column and higher accumulation rates produce a colder ice column. A sentence or two outlining the physics behind these requirements would be useful here.**
We add: "Items 1 and 2 interact, as low accumulation limits the advection of cold, requiring low geothermal heat flow to offset melting. Items 3, 4, and 5 lead to the somewhat contradictory requirement of a flat subglacial mountain."

**Section 2.2:**
**Mention that Candidate A is favorable because it is the largest candidate. The other candidates are only a few grid cells large, and are therefore unreliable. The thermomechanical model used to define the candidates is a continuum model, and therefore cannot be expected to accurately describe features on the grid cell scale. Candidate A is the only candidate that is much larger than the grid size, and is therefore the only candidate that can be considered a robust prediction of the model. This is actually the most powerful argument in favor of Candidate A.**
We add in Section 2.2: "The size of Candidate A compared to the other local candidates also makes it more likely that the Van Liefferinge & Pattyn (2013) model captured basal temperatures correctly."

**P 4 L 6: "...basal ice likely traverses the..."**
**The basal ice traversed the trough in the past, replace with "has likely traversed".**
Done
**P 4 L15 – P 5 L2: "...while in the bottom 500 m, a region of more diffuse englacial scattering is present. This distinct zone of basal ice is also apparent in McCoRDS radar data that operates at a higher frequency." This is a good place to reference Bell et al., 2011. The diffuse englacial scattering is similar to what they described as "valley wall" accretion ice near Dome A.**
Added

**P 7 L21: "We...maintain a strict first return policy."**
**In an area of rough basal topography, there is a good chance that the first return may come from off-nadir bed returns. In fact, this is almost certainly what happened, given the results of Section 5. It might be good to include a sentence here mentioning that this first return policy likely resulted in picking off-nadir returns as the bed, and that you explore this in greater detail in the next section.**
We add: "The first return represents a stable interface to interpret in radar, but has a high likelihood of selecting off nadir echoes in steep topography."

**Section 5: A large amount of the material in this section would be more suitable for the results section.**
We move the cross over analysis to the appendix.

**P 10, L5-12: Why not compute H for this dataset (or for the subset of this dataset within Candidate A)? You could determine how RMS roughness varies as a function of window size, and perhaps use this information to**

say something about the processes responsible for shaping the landscape. This goes to my major comment above.

**P 10, L13: "Figure 6 shows the relationship between RMS deviation at 1600 m length scale..." The axes label of Figure 6 says 800 m length scale.**

Corrected to 800 m

**P 10, L15-16: "A stronger relationship is seen for the focused data than for the pik1 data, primarily due to the larger crossover differences seen in the focused data."**

**First comment: the sentence would be clearer if you said "...seen for the focused data than for the unfocused data..." rather than using code ("pik1").**

**Second comment: The second half of this sentence would be more compelling if you said "primarily due to the geometric arguments given earlier". The crossover differences are larger for the focused data than for the unfocused data because the unfocused data includes off-nadir returns in both the along- track and across-track directions, but the focused data only has off-nadir returns in the across-track direction.**

Added

**Figures**
**The map figures need to have some indication of latitude and longitude.**

The figures are placed in the context of a new Figure 1 with latitude and longitude, and added north arrow to all maps.

**Figure 1:**
**Specify in the caption that the "candidates" refer to the cold-based regions. I was looking for specific dots on the map.**

This has been clarified in the figures

**Figure 2:**
**The 10 m contours are extremely difficult to see in a printout. The inset map of Antarctica would be better suited for figure 1. Overlay the boundary of Candidate A.**

This has been clarified in the figures.

**Figure 3: Add more x-axis labels (say, every 10 km). Put the units (dB) on the colorscale. Indicate the boundaries of Candidate A. Add a note to the top left or bottom left corner of the image indicating the direction to Dome C.**

**It might also be helpful to show the echogram going all the way to Dome C. This will allow the x-axis scale to begin at zero, and (more importantly) it will allow the reader to assess how the continuity of the internal layers in Candidate A compares with the continuity of the internal layers at Dome C. If the echogram is expanded this way, you should also add a vertical line indicating the location of the Dome C ice core (or the closest approach to the core), with a tick indicating the lowest depth from which stratigraphically intact ice was recovered.**

A second profile has been added showing the full area from Candidate A to Concordia.

**Figure 4: It might be helpful to show another set of histograms where the range has been truncated at $\pm100$ m, so that the scale is not distorted by a few large outliers.**

Figure 4 has been deleted

**Figure 5: It is hard to see both the crossovers and the bed elevation, as both have similar color scales. It might be better to have the bed elevation in black and white. Alternately, it might be good to have two separate panels, one showing the bed elevation and one showing the crossovers. Consider merging figure 7 with this figure in that case.**

Figure 5 has been deleted

**Figure 6: It is hard to tell the two colors apart, and most of the figure space is blank white space. Consider splitting foc1 and pik1 into separate subplots. Also, consider using log-log axes to more efficiently use all of**

**the space. In addition, the caption says that the RMS window was 1600 m, while the axis label says it was 800 m.**

This figure has been redrawn to be more clear.

**Figure 7: See my note above about potentially merging this figure with figure 5.**

Figure 7 has been merged into a more discussion focused Figure 5.

**Figure 8: Again, was the length scale 800 m or 1600 m? The MOA background adds nothing to the figure, as it is a uniform gray. Consider using Bedmap2 as the background, with the same grayscale as the new bed elevations.**

Figure 8 has been merged into Figure 5.

---

## Referee Report (RR1)

**Resubmission Review:** "High resolution boundary conditions of an old ice target near Dome C, Antarctica"
Duncan A. Young, et al., *The Cryosphere*

**Major Comments:**

My major concerns with the original manuscript were that the paper needed a discussion section plus more scientific interpretation, and that the authors should own their preference for Candidate A. The authors have addressed both of these concerns thoroughly in this revised manuscript. The results section has been expanded and an extensive discussion has been added. The authors not only own their preference for Candidate A, but they also identify a specific location within Candidate A for additional ground-based surveys in preparation for drill site selection. The new manuscript is a solid piece of scientific work and is suitable for publication in *The Cryosphere* after various minor revisions.

**Minor Comments:**

P1, L2-4: "New ice thickness data derived from an airborne coherent radar sounder was combined with unpublished data that was in part unavailable for earlier compilations, and were able to remove older data with high positional uncertainties."
Put this statement in active voice: "*We* combined new ice thickness data... and *we* were able to remove..."

P1 L18: "...with and approximately 400 ka transition..."
Should be "an" not "and".

P2 L3-4: "(1) low accumulation, to restrict vertical thinning rates and increase temporal resolution..."
The effect of surface accumulation is a bit more complex than that. While it is true that low surface accumulation produces low thinning rates (in steady state), it is also true that low accumulation rates produce very thin annual layers. It is the second effect that generally wins out: low-accumulation ice cores (like EPICA Dome C) generally have course temporal resolution but broad temporal coverage spanning multiple glacial cycles, while high-accumulation ice cores (like WAIS Divide or the Greenland cores) have very fine temporal resolution but poor temporal coverage spanning only the last glacial cycle. It may be more accurate to say that low accumulation increases temporal coverage rather than temporal resolution.

P2 L4-5: "(3) proximity to an ice divide to limit vertical thinning rates..."
Is this a reference to the Raymond effect? The average vertical strain rate of the ice column is equal to a/H, the accumulation rate divided by the ice thickness (assuming steady state and neglecting basal melt). This value is independent of distance from the divide or of any other effects associated with the horizontal flow setting. However, the vertical distribution of strain rate within the ice column can change depending on the ice dynamic setting, even if the average value is constant. The Raymond effect can produce an upwarping of layers underneath an ice divide, effectively trading a rapid initial thinning near the surface for much less thinning near the bed, thereby preserving older layers.
If you are referring to the Raymond effect here, it would be appropriate to reference Raymond [1983]. However, the layers in the echograms shown in this paper do not appear to have a visible Raymond arch, so it is possible that either flow over rugged topography or localized wet-based conditions have destroyed the conditions which give rise to the Raymond effect. This clause could also

be removed and point (3) would still stand on the basis of wanting to minimize disturbances due to lateral flow and of wanting to simplify the altitude history of the surface.

P2 L7-8: "Items 1 and 2 interact, as low accumulation limits the advection of cold, requiring low geothermal heat flow to offset melting."
     Rephrase to more clearly describe the underlying physical processes: "Items 1 and 2 interact, as low accumulation limits the *downward* advection of cold *surface temperatures*, requiring low geothermal heat flow to *prevent* melting." (changes in italics)

P2 L8-9: "Items 3, 4, and 5 lead to the somewhat contradictory requirement of a flat subglacial mountain."
     Only items 4 and 5 are involved in the contradiction.

P4 L1-2: "...implying that an ideal old ice target may require a very flat ice-bed interface..."
     What about roughness along the flow path back towards the dome? The old ice near the bed traversed a trajectory from the dome or divide on the way to its present position, and bed roughness along this trajectory could have induced complex deformation fields that distorted the layers even if the drill site itself has a smooth bed. The early part of the trajectory can probably be discounted because the ice was high in the column and therefore mostly unaffected by basal roughness early in its history. However, the later part of the trajectory may have been subject to complex deformation near the bed. This argues for a consideration not only of the local roughness at a potential ice core site, but also of the roughness along a short trajectory pointing back towards the divide. The ideal location would then be at the downstream terminus of a smooth-bed 'stripe' oriented in the flow direction.

P5 L 14-16: "The size of Candidate A compared to the other local candidates also makes it more likely that the Van Liefferinge and Pattyn (2013) model captured basal temperatures correctly."
     It is the size of Candidate A relative to the grid size that makes it more likely that the model captured temperatures correctly, not the size relative to the other candidates. The fact that the other candidates are small relative to the grid size makes it less likely that the model did a good job for them, and therefore strengthens the argument for A; however, the robustness of Candidate A should be independent of the robustness of the other candidates. A possible rephrasing that navigates this distinction is: "The size of Candidate A compared to the 5 km model grid size makes it more likely that the Van Liefferinge and Pattyn (2013) model captured basal temperatures correctly, while the small size of the other candidates relative to the model grid makes them less reliable."

P5 L31-32: "...bedrock trends are significantly disagree..."
     Remove "are".

P6 L3-4: "...a 15 km offset along-track would be required to reconcile the surface slope structure and Bedmap2 bed elevation data at this location."
     What about an across-track offset? Is one offset intrinsically more likely than the other for older navigation systems?

P6 L15-16: "The identification of subglacial lakes is complicated by variations in englacial attenuation that modifies the strong radar reflection due to an ice-water interface (Carter et al., 2007)."
     A better reference here would be Matsuoka, 2011.

P 8 L8: "...and a second line to constrain better an oblique topographic ridge..."
     "constrain better" should be "better constrain".

P 9 L14-15: "To obtain ice thicknesses, we systematically select a window around the earliest bed return, and then automatically select the best fitting pulse waveform within that window (assumed to be a paraboloid power profile), for both the surface and the bed."

By a "paraboloid power profile", am I right in interpreting this to mean that you assume that the echo power has a Gaussian profile on a linear scale, which becomes a parabola on a logarithmic (dB) scale?

P 10 L 13-14: "Regions with a sustained specularity content greater than 0.2 were classified as subglacial lakes."

What do you mean by "sustained"?

P10 L 19: "...all subglacial lakes that were identified had low hydrostatic gradients (Fig 4)."

This is a very powerful argument supporting the presence of subglacial water, but Figure 4 doesn't really allow us to evaluate the hydraulic gradient of most of the lakes (other than the largest ones, which do indeed appear flat by eye). Some quantification of what is meant by "low hydrostatic gradients", and some quantification of whether or not "all" of the lakes do truly meet that criteria, would be helpful here. This doesn't necessarily need any addition to the figure, a simple statement like "X% of the lakes had a hydraulic gradient less than Y" would suffice.

P 14 L 2-4: "Small scale roughness, at length scales of the line spacing and below, is relevant for three reasons: 1) roughness gives insight into the pathways that basal ice must traverse; 2) roughness may provide information on past ice sheet behavior and basal conditions and 3) roughness is a key control on the uncertainties inherent in profiling radar systems."

I would add a fourth factor: 4) basal roughness forces the overriding ice sheet to develop a complex deformation field in the lower part of the ice column, and this deformation field could disturb stratigraphic continuity of the ice core record.

P 16 L12 "...will not be available for melting on the intervalley regions."

"Peaks". The word for "intervalley regions" is "peaks".

P16 L17: "...observed driven stresses..."

Should be "driving stresses".

P17 L17-18: "However, a trade-off is that maintaining a simple flow path for basal ice in such a rough environment will be difficult, and the mountainous region also induces relatively large driving stresses in the overlying ice."

I would add that it's not simply a matter of constraining the flow path (which can also be complicated by unknown changes in ice sheet configuration in the deep past), but of constraining the deformation that a particle of ice accumulates along that flow path. When the basal topography is complex, the flow field in the lower 20-30% of the ice column should be complex as well. As a particle of ice traverses this flow field it accumulates deformation, potentially distorting stratigraphic continuity. I would recommend adding a sentence here about the importance of accumulated deformation along the flow path of the basal ice.

P18 L27: "The result is that the first return will tend toward the minimum ice thickness within the beam pattern, however the measured thickness at this site will be slightly overestimated."

This statement confused me. If the first return is the minimum ice thickness within the beam pattern, then shouldn't the measured ice thickness be an *under*estimate? At first I thought this was a

simple typo (overestimate vs underestimate), but on re-reading it I realize there is an alternate interpretation that also makes sense: the measured ice thickness is an overestimate of the ice thickness *at the cross-track location of the off-nadir return*. The measured thickness is an overestimate of the true ice thickness *at this off-nadir location* because the radio signal took a diagonal path through the ice to get there rather than a vertical path.

However, the measured thickness will still be an underestimate of the true nadir thickness. In the context of error analysis of profile data, the first return will be biased towards systematically underestimating ice thickness. It is this second sense of measurement bias- the bias relative to the true nadir thickness, rather than bias relative to the thickness at an unknown off-nadir position- that people will think of when they read about underestimated or overestimated ice thickness. I would recommend rephrasing this sentence as, "The result is that the first return will tend toward the minimum ice thickness within the beam pattern, and the measured thickness at this site will be systematically underestimated relative to the true nadir thickness."

**Supplementary Material:**
I was not able to locate a table describing the subglacial lakes in the supplementary material. A table listing the centroid lat/lon, mean ice thickness, and along-track length (plus any other variables the authors think are relevant) for each lake should be provided. This table could also be placed in the main text or appendix instead of the supplementary material, as there are only 40 new lakes.

**Figures:**

Fig 2:
Move the label for Candidate A from the right side of the candidate to the left side. In the current configuration it looks like the label refers to the yellow box.

The statement "Regions of disagreement between Bedmap2 and other dataset is shown by the yellow boxes in all panels" is confusing. After reading the text, it is clear that what you are referring to is a misalignment between trends in the surface slope field and trends in the bedrock topography. Clarify this in the caption.

Fig 3:
The caption states that the color scale is relative power after geometric correction. I interpret this to mean that the effects of geometric spreading on echo strength have been removed. However, the shallow layers are still much brighter than the deep layers. Is this because of attenuation, or is my interpretation of this sentence incorrect? If the whole echogram has been geometrically corrected, I would also expect the noise floor near the bed the bed to feature a color ramp, with brighter speckle below and dimmer speckle above, rather than a uniform black.

Fig 5:
There is a problem with the bed elevation and RMS deviation colorbars. Both of them have rendered with a color gradient across the colorbar in addition to the intended color gradient along the colorbar.

The label for the RMS deviation colorbar would be more instructive if it simply said, "Bed Roughness (m)". The caption can clarify that roughness is defined as the RMS deviation of the bed within an 800 m rolling window.

Fig 6:

This colorbar has the same issue as those in Figure 5.

Fig 8:

The x-axis should have the more straightforward and descriptive label, "Roughness", with the definition of roughness (RMS deviation within 800 m window) in parentheses. As written it reads as if the units are multiples of 800 m.

"The focused data has large outliers in rough terrain, as one direction is actually more correct; for the unfocused data, the crossover is smaller, as both directions are equally wrong." I absolutely love this explanation.

---

## Author Response (AR2)

**I thank the reviewers for their detailed comments (and complements!). I respond to the suggested changes below in bold.**

**Response to Anonymous Reviewer #1**

P.1, L.10 "ice of less than 3000 m" -> "ice thickness of less than 3000 m"

**Done**

P.5, L.18 "shown in green on Fig. 3" I did not find green in this Figure. Perhaps the authors meant Fig. 2c.

**Reference fixed**

P.6, L.7 Again, Fig.3 should be Fig.2c. Figure 3 does not seem a reconnaissance line.

**Reworded to: *"Initial radargrams such as that shown in Fig. 4 (upper) show considerable small scale bed roughness, not captured by Bedmap2."***

P.7, Figure 3 bottom Is "North" projected north or true north?

**To the caption for the upper radargram reworded to "*Geographic south is to the right, Dome C and geographic north is to the left, "* and added to the caption for the lower "*…and orientation is the same as above."***

P.8, L.3 -6 It was hard to understand directions of XY (which is X, which is Y?) and geographical directions (projected or true?).

**Reworded to "*with 110 km long longitudinal to slope 'Y' survey lines at separations of down to 1 km cutting across the ice divide, and ~65 km long transverse to slope 'X' tie lines"***

P.9, bottom line of caption of Fig. 4 "Projection for Easting ... Polar Stereographic" This sentence does not provide readers if they use true directions or not.

**Added "and viewed in the projected Northing plane (lower, looking across the ice divide) and projected Easting plane (upper, looking along the ice divide)."**

P.11, Table 2 and Table 3 I am confused because I did not find consistency between Table 2 and Table 3. I mean, "mean deviation" of two tables should agree with a relative manner in principle. But it seems that they do not agree with each other at all in a relative manner. Am I misunderstanding something?

**Reworded captions to "*mean offset and standard deviation"*, added sentence to the main text: "*In all cases, the standard deviation of the trackline data compared to the OIA-only grid was better than the comparison to Bedmap2 (see Table 3), likely related to the loss of spatial resolution in Bedmap2."***

P.11, footnote "*" of Table 3 It seems that the difference is from Bedmap2 here, and not from the OIA-only grid. Am I misunderstanding something?

**Fixed**

P.12, 3rd and 4th lines in the caption of Fig. 5. The authors wrote, "show that all of the candidates aside from the innermost portion of Candidate A lie over regions of high driving stress". I do not see so on the map. I think this statement needs to be

reconsidered.

**Reworded caption to "...** *show that all of the candidates aside from the innermost portion of Candidate A lie over regions of* relatively *high* (20-30 kPa) *driving stress"*

**Response to Anonymous Referee #3**

Minor Comments:

P1, L2-4: "New ice thickness data derived from an airborne coherent radar sounder was combined with unpublished data that was in part unavailable for earlier compilations, and were able to remove older data with high positional uncertainties."
Put this statement in active voice: "We combined new ice thickness data... and we were able to remove..."

**Done**

P1 L18: "...with and approximately 400 ka transition..." Should be "an" not "and".

**Done**

P2 L3-4: "(1) low accumulation, to restrict vertical thinning rates and increase temporal resolution..." The effect of surface accumulation is a bit more complex than that. While it is true that low...

**Changed "temporal resolution" to "temporal coverage".**

P2 L4-5: "(3) proximity to an ice divide to limit vertical thinning rates..."

**Removed "vertical thinning and"**

P2 L7-8: "Items 1 and 2 interact, as low accumulation limits the advection of cold, requiring low geothermal heat flow to offset melting."
Rephrase to more clearly describe the underlying physical processes: "Items 1 and 2 interact, as low accumulation limits the downward advection of cold surface temperatures, requiring low geothermal heat flow to prevent melting."

**Rewritten as suggested**

P2 L8-9: "Items 3, 4, and 5 lead to the somewhat contradictory requirement of a flat subglacial mountain." Only items 4 and 5 are involved in the contradiction.

**It is the relatively high ice of the ice divide, along with the limited ice thickness that forced the existence of a mountain to satisfy this criteria. Rewritten as follows:** *"Items 3 (*implying elevated ice surface height*), 4 (*smooth subglacial topography*), and 5 (*implying limited ice thickness*) lead to the somewhat contradictory requirement of a flat subglacial mountain. Given the significant logistical requirements of ice core recovery, another important criterion for any old ice site is accessibility."*

P4 L1-2: "...implying that an ideal old ice target may require a very flat ice-bed interface..." What about roughness along the flow path back towards the dome? The old ice near the bed...

**Added "around a flowline tracing back toward the ice divide"**

P5 L 14-16: "The size of Candidate A compared to the other local candidates also makes

it more likely that the Van Liefferinge and Pattyn (2013) model captured basal temperatures correctly."… A possible rephrasing that navigates this distinction is: "The size of Candidate A compared to the 5 km model grid size makes it more likely that the Van Liefferinge and Pattyn (2013) model captured basal temperatures correctly, while the small size of the other candidates relative to the model grid makes them less reliable."

**Reworded to: "The size of Candidate A compared to *the 5 km model cell size*…"**
P5 L31-32: "...bedrock trends are significantly disagree..." Remove "are".

**Done**

P6 L3-4: "...a 15 km offset along-track would be required to reconcile the surface slope structure and Bedmap2 bed elevation data at this location." What about an across-track offset? Is one offset intrinsically more likely than the other for older navigation systems?

**Added the following: "*(as the flight line crosses the trough, the interpolated topography is not sensitive here to cross track errors on this line)*"**

P6 L15-16: "The identification of subglacial lakes is complicated by variations in englacial attenuation that modifies the strong radar reflection due to an ice-water interface (Carter et al., 2007)." A better reference here would be Matsuoka, 2011.

**Added Matsuoka, 2011**

P 8 L8: "...and a second line to constrain better an oblique topographic ridge..." "constrain better" should be "better constrain".

**Done**

P 9 L14-15: "To obtain ice thicknesses, we systematically select a window around the earliest bed return, and then automatically select the best fitting pulse waveform within that window (assumed to be a paraboloid power profile), for both the surface and the bed." By a "paraboloid power profile", am I right in interpreting this to mean that you assume that the echo power has a Gaussian profile on a linear scale, which becomes a parabola on a logarithmic (dB) scale?

**Added to radar processing section: "*and the logarithm of signal power was displayed for manual interpretation.*", and reworded this location to: "*(assumed to be a paraboloid power profile in decibels)*"**

P 10 L 13-14: "Regions with a sustained specularity content greater than 0.2 were classified as subglacial lakes." What do you mean by "sustained"?

**Deleted "sustained"**

P10 L 19: "...all subglacial lakes that were identified had low hydrostatic gradients (Fig 4)."This is a very powerful argument supporting the presence of subglacial water, but Figure 4doesn't really allow us to evaluate the hydraulic gradient of most of the lakes… This doesn't necessarily need any addition to the figure, a simple statement like "X% of the lakes had a hydraulic gradient less than Y" would suffice.

**Added to the Results section under Additional subglacial lakes: "*50% of segments of specular bed that were 1 km or greater in length had hydraulic head***

*gradients less than 0.1%, meeting the criteria for a lake in Carter et al., 2007, and 71% were less than 0.2%.  This result is consistent with flexural support of small gradients around the edges of these small lakes (Carter et al., 2007)."*

P 14 L 2-4: "Small scale roughness, at length scales of the line spacing and below, is relevant for three reasons: 1) roughness gives insight into the pathways that basal ice must traverse; 2) roughness may provide information on past ice sheet behavior and basal conditions and 3) roughness is a key control on the uncertainties inherent in profiling radar systems."

I would add a fourth factor: 4) basal roughness forces the overriding ice sheet to develop a complex deformation field in the lower part of the ice column, and this deformation field could disturb stratigraphic continuity of the ice core record.

**Added suggested text**

P 16 L12 "...will not be available for melting on the intervalley regions." "Peaks". The word for "intervalley regions" is "peaks".

**As peaks suggests a sharp edge, and many of the "intervalley regions" are flat, reworded as follows: "*...for melting on the highs between valleys.*"**

P16 L17: "...observed driven stresses..." Should be "driving stresses".

**Done**

P17 L17-18: "However, a trade-off is that maintaining a simple flow path for basal ice in such a rough environment will be difficult, and the mountainous region also induces relatively large driving stresses in the overlying ice."…  I would recommend adding a sentence here about the importance of accumulated deformation along the flow path of the basal ice.

**Added *"The paths taken by basal ice elements in such an environment may be torturous, and result in stratigraphic complexity."***

P18 L27: "The result is that the first return will tend toward the minimum ice thickness within the beam pattern, however the measured thickness at this site will be slightly overestimated." … I would recommend rephrasing this sentence as, "The result is that the first return will tend toward the minimum ice thickness within the beam pattern, and the measured thickness at this site will be systematically underestimated relative to the true nadir thickness."

As I wish to force the reader to think both in terms of horizontal as well as vertical uncertainty, I reword as follows:

**"The result is that the first return will tend toward the minimum ice thickness within the aircraft beam pattern, however the measured thickness at the site of reflection will be slightly overestimated.  The primary uncertainty will be in the cross-track position of the bed echo.  Alternatively, if it is assumed that the echo is from nadir, the inferred ice thickness will tend to be underestimated."**

Supplementary Material:

I was not able to locate a table describing the subglacial lakes in the supplementary material. A table listing the centroid lat/lon, mean ice thickness, and along-track length (plus any other variables the authors think are relevant) for each lake should be provided. This table could also be placed in the main text or appendix instead of the supplementary material, as there are only 40 new lakes.

**A csv file is now included with the requested parameters, as part of generating this file new lakes were found, changing in the numbers (but not the conclusions) in the results section.  as the number of lakes has increased to 54, I have elected not to put them in the main text.**

Figures:

Fig 2: Move the label for Candidate A from the right side of the candidate to the left side.

**Done**

In the current configuration it looks like the label refers to the yellow box. The statement "Regions of disagreement between Bedmap2 and other dataset is shown by the yellow boxes in all panels" is confusing. After reading the text, it is clear that what you are referring to is a misalignment between trends in the surface slope field and trends in the bedrock topography. Clarify this in the caption.

**Reworded to: "Regions where trends seen in surface data are absent in Bedmap2 are shown by the yellow boxes in all panels."**

Fig 3: The caption states that the color scale is relative power after geometric correction. I interpret this to mean that the effects of geometric spreading on echo strength have been removed. However, the shallow layers are still much brighter than the deep layers. Is this because of attenuation, or is my interpretation of this sentence incorrect? If the whole echogram has been geometrically corrected, I would also expect the noise floor near the bed the bed to feature a color ramp, with brighter speckle below and dimmer speckle above, rather than a uniform black.

**At these ranges, the geometric loss term near the bed is less than a 2 dB per km, limiting any color ramp near the bed.  I quantified the layer brightness and layers at 1000 meters depth are within 5 dB of layers at 2000 m.  Part of the apparent brightness may be due to surface scattering; therefore I add:**

***"Near surface layers have superposed surface scattering."***

Fig 5:

There is a problem with the bed elevation and RMS deviation colorbars. Both of them have rendered with a color gradient across the colorbar in addition to the intended color gradient along the colorbar. The label for the RMS deviation colorbar would be more instructive if it simply said, "Bed Roughness (m)". The caption can clarify that roughness is defined as the RMS deviation of the bed within an 800 m rolling window.

**Done**

Fig 6:

This colorbar has the same issue as those in Figure 5.

**Done**

Fig 8:

The x-axis should have the more straightforward and descriptive label, "Roughness", with the definition of roughness (RMS deviation within 800 m window) in parentheses. As written it reads as if the units are multiples of 800 m.

**Done - plots were replotted with slightly different data**